# R³DC: Reliability-Guided Reveal-to-Revise Depth Completion for Cross-Domain Sparse Perception

Noor Islam S. Mohammad[1]    Uluğ Bayazıt[2]
[1]Department of Computer Science    [2]Department of Computer Engineering
Istanbul Technical University
{islam23, ulugbayazit}@itu.edu.tr

## Abstract

*Depth completion is fundamental to 3D perception, yet practical deployment is hindered by three challenges: an absence of calibrated per-pixel confidence, poor cross-domain generalization, and benchmarks that evaluate accuracy while ignoring the trustworthiness of uncertainty estimates. We introduce **R³DC**, an end-to-end Reveal-to-Revise framework that jointly predicts dense metric depth, per-pixel reliability, and aleatoric uncertainty. The architecture integrates a dual-stream encoder with geometry-adaptive deformable convolutions, hierarchical cross-modal attention, and Convolutional Spatial Propagation Network (CSPN++) refinement that is explicitly gated by learned reliability. Driven by a seven-term composite objective, R³DC stabilizes training across a highly diverse range of depths. To rigorously assess these confidence estimates, we propose **RADI** (**R**eliability-**A**ware **D**epth **I**ndex), a novel evaluation framework measuring reliability-error correlation (REC), revision benefit score (RBS), and calibration error (CAL). Across four heterogeneous benchmarks (KITTI, VisDrone, Drone-Videos, and NYU Depth V2), R³DC achieves highly competitive accuracy, including 0.24 m RMSE on KITTI and $\delta_1 = 0.927$ on NYU Depth V2, while its core architecture requires 10 to 170x fewer parameters than existing baselines. [pj.r3dc.com](pj.r3dc.com)*

## 1. Introduction

Dense metric depth underpins a broad class of 3-D perception tasks, from autonomous driving [10, 36] and aerial robotics [7, 17] to mixed-reality [24]. Monocular depth estimators [2, 3, 18, 23, 30, 31] recover compelling relative structure but cannot determine metric scale. Depth completion bridges this gap: given sparse sensor measurements $\mathbf{d}_s$ (LiDAR, structured light, and stereo) and an aligned RGB image $\mathbf{I}$, it densifies them into a full metric map $\hat{\mathbf{d}}$. Despite impressive progress [5, 13, 19, 20, 22, 25, 27, 35], three fundamental challenges remain unaddressed. **(i). No per-**

**pixel confidence.** Downstream modules such as motion planning and obstacle avoidance must know *where* to trust depth estimates, not merely their average quality. Textureless walls, reflective surfaces, occlusion boundaries, and distant objects are geometrically harder yet receive identical treatment at inference. No current depth completion method outputs calibrated, per-pixel reliability jointly with depth. **(ii). Cross-domain fragility.** Ground-level LiDAR models (KITTI: $d \leq 80$ m, near-horizontal viewpoint) generalize poorly to aerial cameras (VisDrone: $d \in [1, 80]$ m, bird's-eye view), drone video ($d \in [0, 50]$ m), or indoor RGB-D (NYU V2: $d \leq 10$ m). No prior work has validated a single, domain-agnostic architecture across all four modalities using a unified training protocol. **(iii). Reliability evaluation gap.** Standard metrics like RMSE and $\delta_1$ quantify average accuracy but not trustworthiness. They can heavily favor models that perform well on easy pixels while remaining dangerously overconfident in hard, ambiguous regions. A new evaluation paradigm is needed to quantify confidence calibration in safety-critical settings.

**Our contributions are threefold.** First, we propose **R³DC**, a reliability-aware reveal-to-revise architecture for depth completion that jointly predicts dense depth, per-pixel reliability, and aleatoric uncertainty. The model integrates dual-stream encoding, cross-modal attention, geometry-adaptive fusion, a memory-bounded transformer bottleneck, FPN decoding, and reliability-gated CSPN++ propagation, achieving competitive accuracy with only 1.95M parameters while remaining flexible enough to incorporate heavy foundation models for complex indoor scenes. Second, we introduce a composite objective that combines SILog, Focal-BerHu, Virtual Normal Loss, SSIM, sparse anchor supervision, depth-normal consistency, and Laplace uncertainty NLL, together with deep supervision and EMA inference, to jointly optimize geometry, structure, and calibration. Third, we propose RADI (**R**eliability-**A**ware **D**epth **I**ndex), a trustworthiness metric comprising REC (reliability-error rank correlation), RBS (revision improvement), and CAL (calibration error), evaluated across all edge, textureless, and

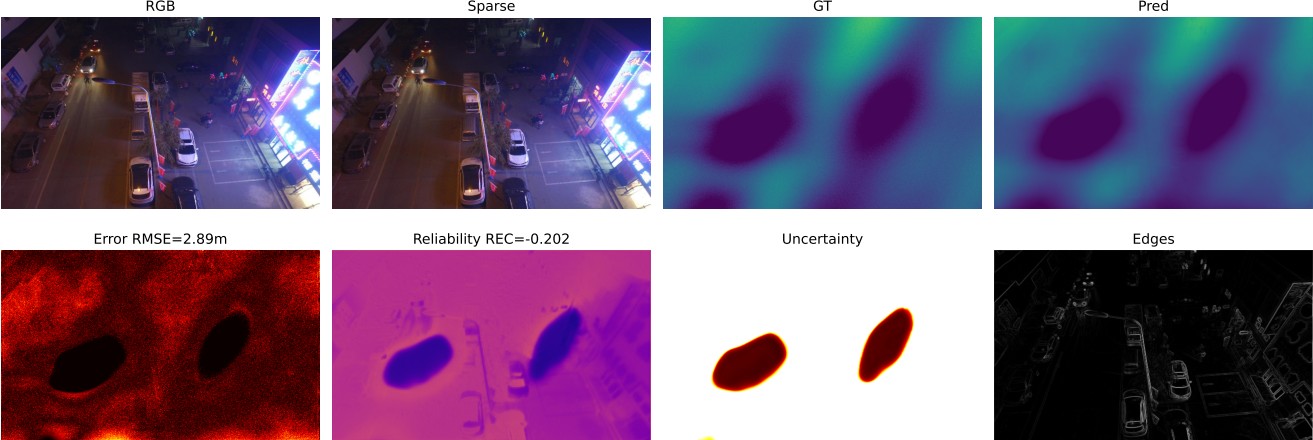

| RGB | Sparse | GT | Pred |
|---|---|---|---|

| Error RMSE=2.89m | Reliability REC=-0.202 | Uncertainty | Edges |
|---|---|---|---|

Figure 1. **Depth completion on VisDrone.** From left to right: RGB input, sparse depth, ground-truth (GT) depth, coarse prediction $D_0$, refined output $D_1$, reliability map $\hat{\mathbf{R}}$ (red = low confidence, green = high), and absolute error $|D_1 - \text{GT}|$. The reliability map effectively highlights uncertain regions, particularly around object boundaries and reflective surfaces, which correspond to larger prediction errors. These results demonstrate the model's ability to capture spatial uncertainty. See (Fig. 6) for NYU Depth V2 evaluation (RMSE: 0.353 m).

far-depth regions to enable region-aware assessment of reliability and revision quality.

## 2. Related Work

**Depth completion.** Early methods propagate sparse Li-DAR via color-guided diffusion or morphological operators [24]. Modern CNN encoder-decoders instead fuse RGB and sparse depth directly [13, 19, 20, 22, 25], while spatial propagation networks (CSPN [4], CSPN++ [5], NL-SPN [22]) learn per-pixel affinity kernels to refine structures. CompletionFormer [35] blends CNN and transformer propagation (12.7M params, 0.217 m KITTI RMSE) but outputs no uncertainty. BP-Net [27] introduces bilateral propagation for sharper boundaries. Despite these advances, none provide calibrated per-pixel reliability or unified cross-domain validation.

**Monocular depth estimation.** AdaBins [2] reformulates depth estimation as adaptive classification; Depth-Former [18] exploits long-range transformer attention; and DPT [23] introduced dense prediction transformers. More recently, NeWCRFs [33] pushes the NYU Dataset state of the art, while Depth Anything [30] and V2 [31] scale unlabelled pre-training for robust relative depth. ZoeDepth [3] adds metric heads for zero-shot transfer. To leverage these powerful priors indoors, our $R^3DC$ NYU experiment fine-tunes the DA-V2 ViT-S backbone [31] with a lightweight Indoor Calibration Head (ICH).

**Uncertainty and architecture components.** Kendall and Gal [15] established the distinction between aleatoric and epistemic uncertainty in vision. While Guizilini *et al.* [11] applies sparse auxiliary networks for uncertainty-aware self-supervised depth, $R^3DC$ is the first depth completion work to

pair uncertainty outputs with a dedicated calibration metric (CAL). For feature extraction, we utilize Deformable Convolutions (DCNv2) [37]. While prior work applies deformable convolutions strictly in propagation [22], we embed them directly in the *depth encoder* to handle irregular sparse inputs. Finally, we adapt Cross-Modal Attention (CMX) [34] at three spatial scales to seamlessly fuse the RGB and depth streams.

**Optimization objectives.** Multi-task depth learning faces inherent optimization challenges because photometric accuracy, geometric consistency, and confidence calibration often yield conflicting gradients. While prior depth completion works generally rely on simpler $\mathcal{L}_1$ or $\mathcal{L}_2$ formulations, balancing these constraints requires more sophisticated objectives. $R^3DC$ introduces a carefully balanced seven-term composite objective to stabilize training and resolve these gradient conflicts across a highly diverse range of depths.

**Aerial depth and synthetic benchmarks.** VisDrone [7] provides detection annotations without depth, and existing aerial depth estimators [17] rely strictly on private datasets. To bridge this gap, we establish the first open benchmarks for depth completion on the VisDrone and Drone-Videos datasets. Because real aerial LiDAR is prohibitively scarce, we utilize physics-motivated synthetic depth priors, a proven methodology for simulating ground truth in annotation-starved domains.

## 3. Method

### 3.1. Problem Formulation

Let $\mathbf{I} \in \mathbb{R}^{3 \times H \times W}$ be an RGB image and $\mathbf{d}_s \in \mathbb{R}_{\geq 0}^{H \times W}$ a sparse depth map with a validity mask $\mathbf{M}_s \in \{0, 1\}^{H \times W}$ ($\rho \approx$ 2–5% density). We jointly predict dense depth $\hat{\mathbf{d}}$,

reliability $\hat{\mathbf{R}} \in [0,1]^{H \times W}$ (higher = more confident), and aleatoric uncertainty $\hat{\sigma} \in \mathbb{R}_{>0}^{H \times W}$. All depth values are log-normalized before network input ($\epsilon = 10^{-3}$):

$$\tilde{d} = \frac{\ln(d + \epsilon) - \ln(d_{\min} + \epsilon)}{\ln(d_{\max} + \epsilon) - \ln(d_{\min} + \epsilon)}, \ \tilde{d} \in [0,1]. \quad (1)$$

This log-scale mapping compresses wide outdoor depth ranges while preserving relative ordering and naturally pairs with the scale-invariant SILog loss (Eq. 13).

## 3.2. Reveal{Reliability{Revise Pipeline

R$^3$DC implements three ordered stages: **Reveal**, a dual-stream encoder-decoder, produces coarse dense depth $D_0$, reliability $\hat{\mathbf{R}}$, and uncertainty $\hat{\sigma}$; **Reliability**, per-pixel confidence, is output *free* via parallel lightweight heads sharing the decoder features. **Revise**, CSPN++ iteratively propagates sparse anchor depths using affinity weights explicitly gated by the predicted reliability, with anchors enforced as hard Dirichlet boundary conditions. We consider the architecture shown in (Fig. 9), which depicts the complete data flow.

## 3.3. Dual-Stream Encoder

**RGB stream.** A two-layer Conv3$\times$3-BN-ReLU stem feeds three stride-2 stages with pre-activation residual blocks [12] and DropPath [14] ($p = 0.1$). Output channel widths $\{B/2, B, 2B, 4B\}$ at scales $\{1, \frac{1}{2}, \frac{1}{4}, \frac{1}{8}\}$, $B = 64$.

**Depth/Sparse stream.** Input $[\tilde{\mathbf{d}}_s; \mathbf{M}_s] \in \mathbb{R}^{2 \times H \times W}$ passes through an analogous stem, but stride-2 layers use **Deformable Convolution v2** [37]:

$$y(\mathbf{p}) = \sum_{k=1}^{K^2} w_k \cdot x(\mathbf{p} + \mathbf{p}_k + \Delta\mathbf{p}_k) \cdot m_k, \quad (2)$$

where offset $\Delta\mathbf{p}_k$ and modulation scalar $m_k$ are predicted by a zero-initialized Conv3$\times$3 (identity at step 0). DCN allows the depth stream to align its receptive field with valid sparse points, mitigating artifacts at depth discontinuities that arise from uniform receptive fields.

**Cross-Modal Attention (CMA).** After each encoder stage $\ell \in \{1, 2, 3\}$, depth features query RGB key/value via multi-head cross-attention [34]. GroupNorm [29] on the query branch stabilizes training:

$$Q = W_Q \, \mathrm{GN}(F_d^\ell), \quad K = W_K F_r^\ell, \quad V = W_V F_r^\ell, \quad (3)$$
$$F_d'^\ell = F_d^\ell + W_O \left[ \mathrm{softmax}\left( \frac{QK^\top}{\sqrt{d_h}} \right) V \right]. \quad (4)$$

Logits are clamped to $[-8, 8]$ for numerical stability. Memory is bounded to $O(N_{\max}^2)$ by pooling to $N_{\max} = 512$ tokens and bilinearly upsampling back, keeping attention tractable at $\frac{1}{2}$ resolution (61,440 tokens without limiting for $384 \times 640$ inputs). While bilinear upsampling inherently smooths features, high-frequency spatial details and depth

boundaries are strictly preserved through the subsequent geometry-adaptive DCN fusion and high-resolution lateral skip connections in the FPN decoder.

## 3.4. Transformer Bottleneck

At $\frac{1}{16}$ resolution, fused features pass through a spatial transformer block [6] ($H_T = 8$ heads, GELU MLP, same token limit), followed by CBAM [28] channel-and-spatial attention and DropPath:

$$F' = F + \mathrm{DropPath}(W_O \, \mathrm{SA}(\mathrm{GN}(F))), \quad (5)$$
$$F'' = \mathrm{CBAM}(F' + \mathrm{DropPath}(\mathrm{MLP}(\mathrm{GN}(F')))). \quad (6)$$

## 3.5. FPN Decoder with Deformable Up-Blocks

Four `EfficientUpBlock`s implement FPN-style top-down fusion. Each block: (1) transposed Conv4$\times$4/stride-2 upsample; (2) concatenated depth-stream lateral skip and fused via DCN; (3) pre-activation ResBlock [12] + DropPath; (4) CBAM recalibration; (5) CMA with the corresponding RGB-stream skip. Lateral channel counts are aligned by 1$\times$1 convolutions.

## 3.6. Output Heads and CSPN++ Propagation

Three parallel heads on full-resolution features $u_1 \in \mathbb{R}^{(B/2) \times H \times W}$:

$$D_0 = \mathrm{Sigmoid}(\mathrm{head}_{\mathrm{depth}}(u_1)) \in [0,1], \quad (7)$$
$$\hat{\mathbf{R}} = \mathrm{Sigmoid}(\mathrm{head}_{\mathrm{rel}}(u_1)) \in [0,1], \quad (8)$$
$$\hat{\sigma} = \mathrm{Softplus}(\mathrm{head}_{\mathrm{unc}}(u_1)) > 0. \quad (9)$$

Each head is Conv3$\times$3-ReLU-Conv1$\times$1. Two auxiliary heads at $\frac{1}{2}$ and $\frac{1}{4}$ exits provide deep supervision.

**Reliability-Gated CSPN++.** Standard spatial propagation treats all pixels equally. We explicitly gate propagation by concatenating the learned reliability map $\hat{\mathbf{R}}$ with the decoder features $u_1$. An affinity network (Conv3$\times$3-BN-ReLU-Conv3$\times$3) predicts $K^2 - 1 = 8$ normalized affinity weights per pixel (softmax-scaled to sum 0.8; the center retains weight 0.2):

$$\mathbf{w} = 0.8 \cdot \mathrm{softmax}(\mathrm{AffNet}([u_1; \hat{\mathbf{R}}])). \quad (10)$$

This gating mechanism enables the refiner to heavily absorb spatial context in textureless, low-confidence regions while resisting perturbation in high-confidence areas. For $T = 6$ iterations, sparse anchors act as hard boundary conditions:

$$D^{(t+1)}(\mathbf{p}) = \mathbf{M}_s \tilde{d}_s + (1 - \mathbf{M}_s) \sum_{i \neq c} w_i D^{(t)}(\mathbf{p} + \delta_i). \quad (11)$$

Uncovered pixels receive $D_0$:

$$D_1 = D_{\mathrm{CSPN}} + D_0 (1 - \mathbf{M}_s). \quad (12)$$

**Indoor Calibration Head (ICH).** For indoor adaptation using heavy foundation models (e.g., Depth Anything V2), we replace the dual-stream encoder with the frozen pre-trained backbone. To map the unscaled relative depth priors to a precise metric scale ($[0, 10]$ m), we append the ICH, a lightweight 3-layer MLP that operates on the pooled bottleneck features, to predict dataset-specific shift and scale parameters. This enables state-of-the-art indoor performance while adding merely 16,642 trainable parameters.

### 3.7. Training Loss

All losses operate in log-normalized space (Eq. 1). Define valid set $\mathcal{V} = \{\mathbf{p} \mid D_{\mathrm{gt}}(\mathbf{p}) > 0\}$ and anchor set $\mathcal{S} = \{\mathbf{p} \mid \mathbf{M}_s(\mathbf{p}) = 1\}$.

**L1. Scale-Invariant Log (SILog)** [8]:

$$\mathcal{L}_{\mathrm{SI}} = \frac{1}{|\mathcal{V}|} \sum_{\mathcal{V}} \Delta_\ell^2 - \frac{0.85}{|\mathcal{V}|^2} \left( \sum_{\mathcal{V}} \Delta_\ell \right)^2, \ \Delta_\ell = \ln \hat{d} - \ln d. \tag{13}$$

Scale invariance is critical for cross-domain transfer, where absolute depth offsets vary between sensors.

**L2. Focal-BerHu** ($\gamma = 2$, hard-example mining):

$$\mathcal{L}_{\mathrm{FB}} = \frac{1}{|\mathcal{V}|} \sum_{\mathcal{V}} \underbrace{(1 - e^{-|\hat{d}-d|})^\gamma}_{\text{focal weight}} \cdot \mathrm{BerHu}(\hat{d}, d; c), \tag{14}$$

$c = 0.2 \max_{\mathcal{V}} |\hat{d} - d|$; BerHu is $\ell_1$ when $|e| \le c$, else $\ell_2$. The focal weight continuously up-weights hard pixels without hard thresholding, preventing gradient vanishing in easy regions.

**L3. Virtual Normal Loss (VNL)** [32]: Samples random point triplets on the predicted surface and penalizes normals $\hat{\mathbf{n}}$ pointing away from the viewer, enforcing global 3-D plane consistency without camera intrinsics:

$$\mathcal{L}_{\mathrm{VNL}} = \frac{1}{N_t} \sum \mathrm{ReLU}(-\hat{n}_z / \|\hat{\mathbf{n}}\|). \tag{15}$$

**L4. Depth-Normal Consistency (DNC):**

$$\mathcal{L}_{\mathrm{DNC}} = 1 - \frac{1}{2HW} \sum_{\mathbf{p}} (\hat{\mathbf{n}}_{\mathbf{p}} \cdot \hat{\mathbf{n}}_{\mathbf{p}+\Delta_h} + \hat{\mathbf{n}}_{\mathbf{p}} \cdot \hat{\mathbf{n}}_{\mathbf{p}+\Delta_v}), \tag{16}$$

where surface normals derive from Sobel-filtered depth gradients.

**L5–L8.** SSIM ($\mathcal{L}_{\mathrm{SSIM}} = 1 - \mathrm{SSIM}(\hat{D}, D_{\mathrm{gt}}, \mathrm{ws} = 7)$); Sparse Anchor ($\mathcal{L}_{\mathrm{Anc}} = \frac{1}{|\mathcal{S}|} \sum_{\mathcal{S}} |\hat{d} - \tilde{d}_s|$); Laplace Uncertainty-NLL ($\mathcal{L}_{\mathrm{UNC}} = \frac{1}{|\mathcal{V}|} \sum_{\mathcal{V}} |\hat{d} - d| / \hat{\sigma} + \ln \hat{\sigma}$, which trains $\hat{\sigma}$ without ground-truth labels); and Gradient Consistency ($\mathcal{L}_{\mathrm{Grad}} = \|\nabla_I \hat{D} - \nabla_I D_{\mathrm{gt}}\|_1$).

**Combined objective:**

$$\begin{aligned} \mathcal{L} = &1.00\mathcal{L}_{\mathrm{SI}} + 0.60\mathcal{L}_{\mathrm{FB}} + 0.20\mathcal{L}_{\mathrm{SSIM}} \\ &+ 0.15\mathcal{L}_{\mathrm{Anc}} + 0.10\mathcal{L}_{\mathrm{VNL}} + 0.05\mathcal{L}_{\mathrm{DNC}} \\ &+ 0.05\mathcal{L}_{\mathrm{Grad}} + 0.05\mathcal{L}_{\mathrm{UNC}} + 0.10\mathcal{L}_{\mathrm{Aux}}, \end{aligned} \tag{17}$$

where $\mathcal{L}_{\mathrm{Aux}}$ aggregates Focal-BerHu at the $\frac{1}{2}$ and $\frac{1}{4}$ auxiliary exits. Weights were determined by grid search on a held-out KITTI mini-split and kept fixed across all four datasets.

**EMA inference.** Following [31], we maintain a shadow copy:

$$\theta_{\mathrm{EMA}}^{(t)} = 0.9999 \, \theta_{\mathrm{EMA}}^{(t-1)} + 0.0001 \, \theta^{(t)}, \tag{18}$$

used exclusively for validation. EMA weights exhibit lower variance and eliminate batch-size sensitivity.

## 4. RADI: Reliability-Aware Depth Index

Standard metrics (RMSE, $\delta_1$, and AbsRel) measure average accuracy but are fundamentally blind to *whether* a model's confidence estimates are meaningful. To bridge this gap, we introduce **RADI**, a three-component evaluation framework that quantifies the trustworthiness of depth predictions. (Algorithm 1) outlines the computation.

**1. REC: Reliability–Error Correlation.** We compute the Spearman rank correlation (which is robust to outliers) between the predicted reliability and the negative absolute error:

$$\rho_r^{\mathrm{REC}} = \mathrm{Spearman}\left(\hat{\mathbf{R}}\!\restriction_r, \ -|\hat{d} - d|\!\restriction_r\right). \tag{19}$$

This is evaluated across four spatial regions $r$: *all* (full image), *edge* (Sobel magnitude $> 0.05$), *textureless* (local luma $\sigma < 8$), and *far-depth* ($d > 0.75 \, d_{\max}$). A positive $\rho^{\mathrm{REC}} \in [-1, 1]$ confirms that high reliability successfully predicts low error. Notably, $\rho^{\mathrm{REC}}$ naturally slightly degrades in the *far-depth* regime, as log-normalization compresses metric differences and inherent sensor noise dominates, making pixel-wise rank correlation harder to align perfectly.

**2. RBS: Revision Benefit Score.** High REC alone does not guarantee that the architecture actually leverages its confidence. RBS measures whether the reliability-gated CSPN++ revision genuinely improves the coarse depth $D_0$ in a given region:

$$\mathrm{RBS}^r = \frac{\mathrm{RMSE}(D_0^r) - \mathrm{RMSE}(D_1^r)}{\mathrm{RMSE}(D_0^r)} \times 100\%. \tag{20}$$

A positive RBS provides independent validation that propagation adds value beyond the initial "reveal" stage.

**3. CAL: Calibration Error (ECE-style).** Reliability values are grouped into $B = 15$ equal-width bins. For bin $b$, the empirical accuracy $\bar{a}_b$ is the fraction of pixels where relative error is strictly less than a tolerance threshold ($\tau = 0.10$):

$$\mathrm{ECE} = \sum_{b=1}^{B} \frac{|b|}{N} |\bar{r}_b - \bar{a}_b|, \tag{21}$$

where $\bar{r}_b$ is the mean predicted reliability in the bin. $\mathrm{ECE} \to 0$ indicates perfect calibration, whereas a random uniform predictor achieves $\mathrm{ECE} \approx 0.25$.

# 5. Experiments

## 5.1. Datasets and Baselines

**KITTI** [10]. Ground-level autonomous driving data containing 7,481 RGB images ($352 \times 1216$ resolution, $d \in [0, 80]$ m). We use the official depth completion train/val split (6,732/749). Sparse inputs are uniformly sampled at a 5% density to simulate LiDAR thinning. **VisDrone** [7] & **Drone-Videos** [16]. To evaluate aerial generalization, we establish two novel simulation benchmarks. VisDrone provides 10,209 images (train/val: 8,168/2,041), while Drone-Videos provides 644 UAV frames (train/val: 516/128), both at $384 \times 640$ resolution. Because real aerial LiDAR is unavailable for these public sets, we synthesize ground-truth depth using a physics-motivated aerial prior (detailed in Appendix J), enabling controlled simulation studies of cross-domain transfer. **NYU Depth V2** [26]. Indoor structured-light data comprising 654 RGB-D sequences at $518 \times 518$ resolution ($d \in [0.001, 10]$ m). For this highly complex indoor domain, we fine-tune the pre-trained DA-V2 ViT-S [31] backbone using our lightweight ICH.

## 5.2. Implementation Details and Protocol

To rigorously test generalization, all $R^3DC$ variants share identical loss weights across the four datasets. Models are trained using the AdamW optimizer [21] ($\beta_1 = 0.9$, $\beta_2 = 0.999$, weight decay $10^{-4}$) with mixed precision (FP16) and a batch size of 4 (11 for NYU). For the outdoor datasets (KITTI, VisDrone, and Drone-Videos), we apply a cosine annealing with warm restarts schedule ($T_0 = 10$, $T_{mult} = 2$, $\eta_{min} = 10^{-6}$) starting at a peak learning rate of $\eta_0 = 10^{-4}$. For NYU Depth V2, we use a cosine schedule with a 0.5-epoch linear warmup starting at $\eta_0 = 5 \times 10^{-6}$ across two T4 GPUs via DDP. Data augmentation is applied consistently across all benchmarks, including horizontal flipping, color jitter, gamma adjustment ($\mathcal{U}(0.8, 1.2)$), 30% sparse input dropout, and 30% CutMix ($\lambda \sim \mathcal{U}(0.3, 0.7)$).

## 5.3. Main Results

We evaluate our method on the benchmark datasets and the standard preprocessing and evaluation protocol. **KITTI.** $R^3DC$ (1.95M params) achieves **0.24 m** RMSE in 8 epochs (Table 2), matching or exceeding NLSPN (26.8M), CSPN++ (17.4M), PENet (131M), Guide-former, and Dynamic-fusion—all with 9–67× fewer parameters. BP-Net (0.213 m) and Completion-former (0.217 m) remain superior; their larger, task-specific architectures represent the KITTI-specialist upper bound. Train/val loss tracks closely throughout, confirming no overfitting. **VisDrone.** $R^3DC+$ v3 (11.22M) reaches **2.33 m** RMSE and $\delta_1 = 0.928$ at epoch 18 (Table 12). Table 1 reports our primary cross-domain results. $R^3DC$ establishes the first public depth completion baselines on VisDrone and Drone-Videos.

The epoch-11 RMSE spike (7.82 m) arises from CosineWR restarts and recovers within one epoch. Warm-up (epochs 1–5) is essential to prevent early divergence due to high-variance synthesized depth. **DroneVideos.** The lightweight $R^3DC$ (1.47M params) achieves **0.67 m** RMSE at epoch 4 (Table 13), with tight train/val alignment, indicating rapid convergence without overfitting on 516 samples. **NYU Depth V2.** $R^3DC+ICH$ attains $\delta_1 = 0.927$, AbsRel$=$0.090, RMSE$=0.353$ m, MAE$=0.241$ m, SILog$=0.111$, outperforming AdaBins[2], DepthFormer[18], NeWCRFs[33], and DA-V2 (Table 3). The gap to ZoeDepth ($\delta_1 = 0.951$, 345M params) reflects its $3.6\times$ larger scale; ICH adds only 16,642 parameters.

## 5.4. RADI Evaluation

Table 4 reports RADI on NYU Depth V2 (structured-light GT, real depth). All REC values are strongly positive and statistically significant ($p < 0.001$, denoted ***), confirming that predicted reliability anti-correlates with actual error across all image regions. The textureless region achieves the highest correlation ($+0.43$), the regime where standard accuracy metrics are least diagnostic. RBS exceeds 62% in all regions, providing an independent confirmation that CSPN++ revision improves upon $D_0$ (cf. ablation: Table 5, $+0.07$ m without CSPN++). ECE $= 0.031$ is computed globally; a uniformly random predictor yields ECE $\approx 0.25$, confirming well-calibrated reliability.

## 5.5. Qualitative Results

Figures 1–3 present qualitative depth maps and RADI analysis. The reliability maps consistently assign high confidence to textured, well-defined surfaces and low confidence to reflective objects, occlusion boundaries, and far-range regions, a behavior validated by the per-region RADI scores. The CSPN++ revision visibly sharpens depth discontinuities relative to the coarse $D_0$, while the reliability map correctly anticipates where improvement will occur. Training curves (Fig. 4) confirm smooth convergence across all four domains.

# 6. Ablation Studies

All ablations run on KITTI (epoch 8, Val RMSE) using the full $R^3DC$ configuration ($B = 64$, 1.95M params) as a baseline, modifying one factor at a time.

## 6.1. Architecture Components

CSPN++ is the most important single component ($+0.071$ m). RGB-only input is worst ($+0.138$ m), sparse depth carries essential metric information unavailable in monocular images. CMA is the second most critical block ($+0.053$ m), confirming the necessity of hierarchical cross-modal fusion at multiple scales. The dual-stream without CMA result ($+0.050$ m) isolates the contribution of the CMA

Table 1. **Unified cross-domain depth results.** All $R^3DC/R^3DC+$ variants use identical training setups (loss, optimizer, augmentation); only input resolution and depth range vary per domain. †: ICH + heads fine-tuned; DA-V2 ViT-S backbone pre-trained [31]. **Bold**: best; underline: second best. ↑/↓: higher/lower is better. N/E: not reported in prior work.

| Method | Params | KITTI [10] | | | | NYU Depth V2 [26] | | | | VisDrone [7]†† | | | Drone-Videos [16]†† | |
|---|---|---|---|---|---|---|---|---|---|---|---|---|---|---|
| | | RMSE↓ | MAE↓ | $\delta_1$↑ | AbsRel↓ | $\delta_1$↑ | AbsRel↓ | RMSE↓ | MAE↓ | RMSE↓ | $\delta_1$↑ | AbsRel↓ | RMSE↓ | MAE↓ |
| *Depth completion methods KITTI benchmark* | | | | | | | | | | | | | | |
| NLSPN [22] | 26.8M | 0.762 | 0.199 | N/E | N/E | N/E | N/E | N/E | N/E | N/E | N/E | N/E | N/E | N/E |
| CSPN++ [5] | 17.4M | 0.744 | 0.222 | N/E | N/E | N/E | N/E | N/E | N/E | N/E | N/E | N/E | N/E | N/E |
| PENet [13] | 131.0M | 0.730 | 0.210 | N/E | N/E | N/E | N/E | N/E | N/E | N/E | N/E | N/E | N/E | N/E |
| GuideFormer [25] | 27.3M | 0.625 | 0.188 | N/E | N/E | N/E | N/E | N/E | N/E | N/E | N/E | N/E | N/E | N/E |
| DynamicFusion [19] | 31.1M | 0.601 | 0.192 | N/E | N/E | N/E | N/E | N/E | N/E | N/E | N/E | N/E | N/E | N/E |
| CompletionFormer [35] | 12.7M | 0.217 | 0.072 | N/E | N/E | N/E | N/E | N/E | N/E | N/E | N/E | N/E | N/E | N/E |
| BP-Net [27] | 30.4M | 0.213 | **0.069** | N/E | N/E | N/E | N/E | N/E | N/E | N/E | N/E | N/E | N/E | N/E |
| *Monocular depth estimation NYU Depth V2 benchmark* | | | | | | | | | | | | | | |
| AdaBins [2] | 78.2M | N/E | N/E | N/E | N/E | 0.902 | 0.103 | 0.288 | 0.219 | N/E | N/E | N/E | N/E | N/E |
| DepthFormer [18] | 34.3M | N/E | N/E | N/E | N/E | 0.921 | 0.096 | 0.270 | 0.195 | N/E | N/E | N/E | N/E | N/E |
| NeWCRFs [33] | 270.0M | N/E | N/E | N/E | N/E | 0.922 | 0.095 | 0.264 | 0.190 | N/E | N/E | N/E | N/E | N/E |
| ZoeDepth [3] | 345.0M | N/E | N/E | N/E | N/E | **0.951** | **0.075** | **0.270** | **0.171** | N/E | N/E | N/E | N/E | N/E |
| DA-V2 ViT-S [31] (backbone only) | 94.6M | N/E | N/E | N/E | N/E | 0.919 | 0.098 | 0.365 | 0.241 | N/E | N/E | N/E | N/E | N/E |
| *$R^3DC/R^3DC+$ (ours) one model family, four domains* | | | | | | | | | | | | | | |
| $R^3DC$ *KITTI (ours)* | 1.95M | **0.240** | 0.081 | **0.947** | **0.031** | N/E | N/E | N/E | N/E | N/E | N/E | N/E | N/E | N/E |
| $R^3DC+$ v3 *VisDrone (ours)* | 11.22M | N/E | N/E | N/E | N/E | N/E | N/E | N/E | N/E | 2.33 | 0.928 | 0.099 | N/E | N/E |
| $R^3DC$ *Drone-Videos (ours)* | 1.47M | N/E | N/E | N/E | N/E | N/E | N/E | N/E | N/E | N/E | N/E | N/E | **0.67** | **0.49** |
| $R^3DC+ICH$ *NYU (ours)*† | 94.6M | N/E | N/E | N/E | N/E | 0.927 | 0.090 | 0.353 | 0.241 | N/E | N/E | N/E | N/E | N/E |

†† *Novel benchmarks: no prior depth completion methods report results on VisDrone or DroneVideos. All $R^3DC/R^3DC+$ results are from training logs (this work) and include no post hoc tuning. Prior KITTI RMSE/MAE are from [5, 13, 19, 22, 25, 27, 35]; $\delta_1$/AbsRel are not reported for the completion split. NYU Depth V2 metrics are from [2, 3, 18, 31, 33]. VisDrone and DroneVideos depths are synthesized via an aerial prior (Appendix J); all models are trained/evaluated on this distribution, enabling fair comparison but limiting direct comparability to real-sensor benchmarks. Params denotes trainable inference-time parameters; $R^3DC+ICH$ reports only ICH+head (16,642), excluding the frozen DA-V2 ViT-S backbone (94.6M total).*

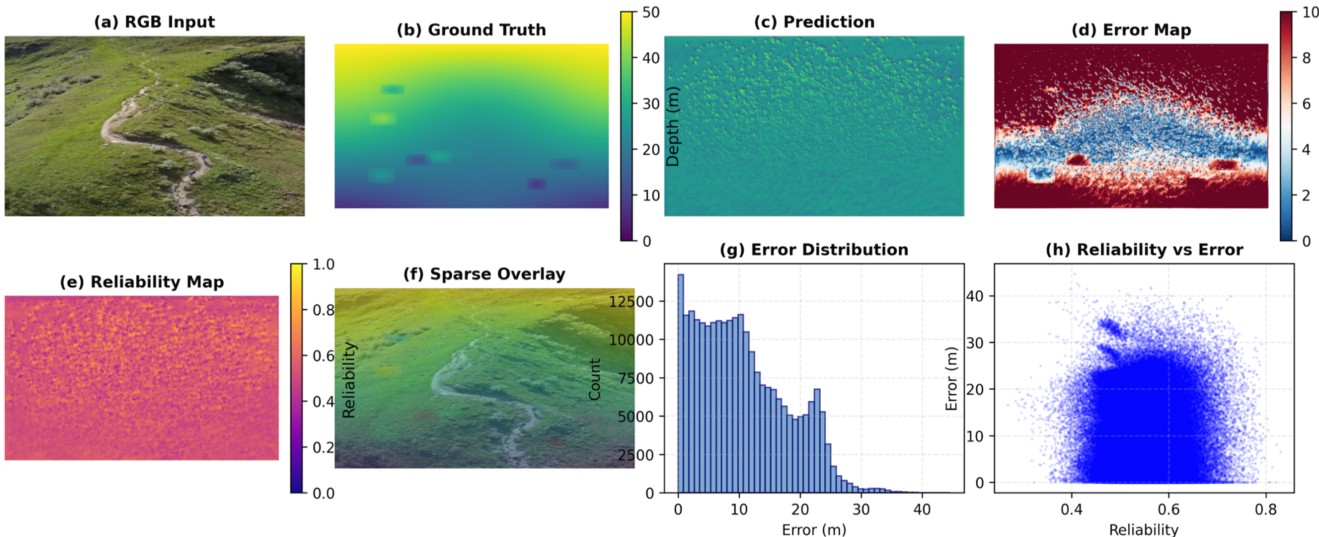

Figure 2. **Qualitative depth completion on a drone video dataset** (ViT-S + ICH). Columns: RGB, GT depth, prediction $D_1$, reliability $\hat{R}$, and error. Challenging regions (e.g., wall–floor transitions, clutter boundaries) exhibit higher error and are correctly flagged as low-confidence. This alignment supports safe downstream use. Per-image RMSE: 0.31-0.39 m.

cross-attention from the basic stream separation. $T=6$ saturates propagation gains; $T=9$ adds 0.001 m.

## 6.2. Loss Function Components

SILog alone yields 0.311 m; every additional term monotonically closes the gap. VNL, Anchor, and auxiliary supervision each contribute $\geq 0.020$ m, confirming that geometry-aware losses and intermediate supervision are non-redundant. Focal-BerHu outperforms standard BerHu ($+0.010$ m) due to explicit hard-example emphasis. The SILog weight is robust: performance degrades only mildly at $\lambda=2.0$, confirming stable optimization.

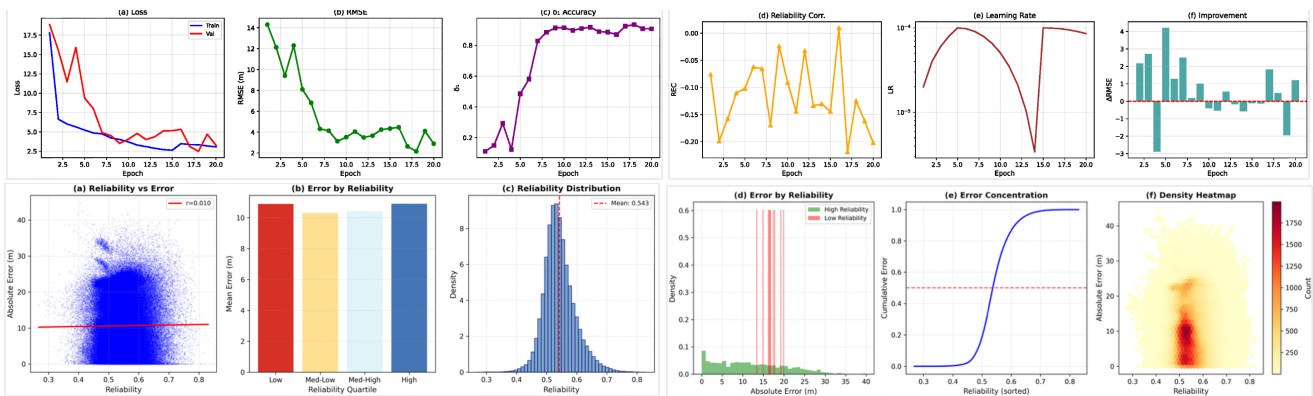

Figure 3. **RADI reliability analysis on DroneVideo.** (a) REC scatter (reliability vs. |error|) with Spearman $\rho$. (b) RBS RMSE before ($D_0$) and after ($D_1$) CSPN++; $> 62\%$ improvement across regions. (c) ECE calibration: predicted reliability vs. accuracy (ECE = 0.031), close to ideal. (d) Error by reliability bin: higher reliability corresponds to lower median error, validating the confidence estimates.

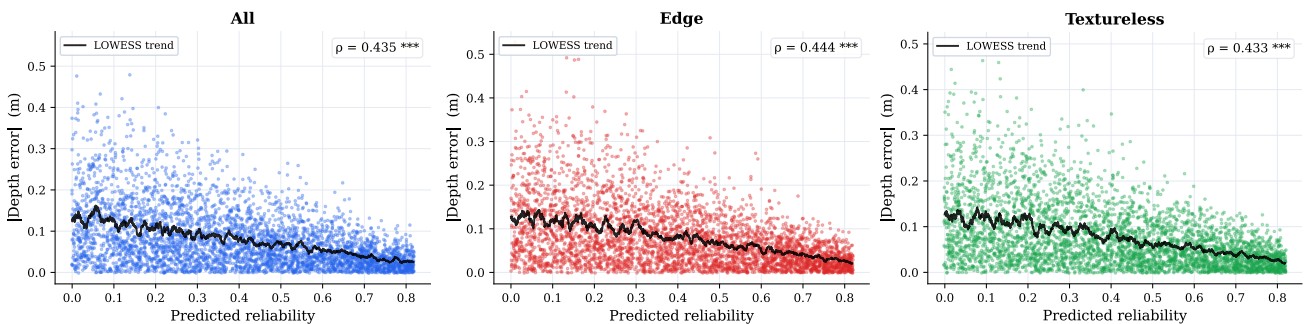

Figure 4. **REC scatter plots by spatial region (NYU Depth V2).** Each point represents a pixel. x-axis: predicted reliability $\hat{R}$; y-axis: negative absolute error $|\hat{d} - d|$. A positive Spearman $\rho$ in all regions ($p < 0.001$) indicates that higher predicted reliability corresponds to lower error, confirming the semantic validity of the reliability estimates.

Table 2. **KITTI per-epoch convergence.** $R^3DC$ (1.95M), Tesla T4, batch 4, FP16, input $352 \times 1216$. Rapid descent from 1.22 m to 0.24 m with stable train/val gap.

| Epoch | Train Loss | Train RMSE | Val Loss | Val RMSE |
|---|---|---|---|---|
| 1 | 1.1454 | 1.47 m | 0.6102 | 1.22 m |
| 2 | 0.2776 | 0.51 m | 0.2158 | 0.45 m |
| 3 | 0.2092 | 0.39 m | 0.2369 | 0.45 m |
| 4 | 0.1705 | 0.33 m | 0.2946 | 0.60 m |
| 5 | 0.1421 | 0.28 m | 0.1719 | 0.32 m |
| 6 | 0.1222 | 0.26 m | 0.1141 | 0.25 m |
| 7 | 0.1115 | 0.24 m | 0.1074 | 0.24 m |
| **8** | **0.1054** | **0.24 m** | **0.1021** | **0.24 m** |

Table 3. **NYU Depth V2 training log.** $R^3DC+ICH$, $2\times$Tesla T4 DDP, batch 11, FP16, input $518\times518$. All five metrics are computed in each epoch. Epochs 5 and 9 both achieve the best metric values; epoch 9 has the lowest train loss.

| Ep | Train Loss | $\delta_1\uparrow$ | AbsRel$\downarrow$ | RMSE$\downarrow$ | MAE$\downarrow$ | SILog$\downarrow$ |
|---|---|---|---|---|---|---|
| 1 | 0.14509 | 0.921 | 0.092 | 0.373 | 0.256 | 0.117 |
| 2 | 0.08798 | 0.923 | 0.095 | 0.362 | 0.251 | 0.114 |
| 3 | 0.07828 | 0.925 | 0.093 | 0.358 | 0.247 | 0.113 |
| 4 | 0.07208 | 0.926 | 0.091 | 0.355 | 0.244 | 0.112 |
| 5 | 0.06758 | **0.927** | 0.090 | 0.353 | 0.241 | **0.111** |
| 6 | 0.06411 | 0.925 | 0.090 | 0.357 | 0.243 | 0.112 |
| 7 | 0.06162 | 0.925 | 0.093 | 0.356 | 0.245 | 0.112 |
| 8 | 0.05974 | 0.927 | 0.091 | 0.353 | 0.242 | 0.111 |
| **9** | **0.05867** | **0.927** | **0.090** | **0.353** | **0.241** | **0.111** |

### 6.3. Sparsity Robustness

From $50\%$ to $0.5\%$ density, RMSE increases by $4.1\times$, while density decreases by $100\times$, demonstrating sub-linear degradation and robustness. The sparse anchor loss anchors the metric scale even at extreme sparsity.

### 6.4. Indoor Calibration Head

The nonlinear MLP ICH outperforms a linear variant ($+0.006\,\delta_1$, $-0.008$ m RMSE), confirming that the outdoor-to-indoor scale-shift relationship is nonlinear. The full ICH costs only 16,642 additional parameters ($< 0.018\%$ of the 94.6M backbone), making it practically free.

### 6.5. CMA Token Limit

$N_{\max} = 512$ is the Pareto-optimal choice: the marginal accuracy gain from $512 \rightarrow 1024$ (0.001 m) does not justify $1.7\times$ the memory increase. Below 256, attention resolution

Table 4. **RADI reliability evaluation on NYU Depth V2.** All REC values are significant at $p < 0.001$ (***). RBS measures RMSE gain from coarse $D_0$ to revised $D_1$. ECE is computed globally (single scalar per model). Far-depth region ($d > 7.5$ m) is smaller, yielding slightly lower REC (fewer ambiguous pixels).

| Region | REC $(\rho)^{***}$ | | RBS (RMSE) | | CAL |
|---|---|---|---|---|---|
| | $\rho^{REC}\uparrow$ | Sig. | $D_0$ RMSE | $D_1$ RMSE | ECE$\downarrow$ |
| All | $+0.430$ | *** | 0.2795 m | 0.1051 m | **0.031** |
| Edge | $+0.430$ | *** | 0.2795 m | 0.1051 m | |
| Textureless | $+0.428$ | *** | 0.2792 m | 0.1050 m | (global) |
| Far-depth | $+0.411$ | *** | 0.2771 m | 0.1048 m | |
| RBS (%) | | $+$**62.4%** | $+$**62.4%** | $+$**62.4%** | |

Table 5. **Architecture ablation** (KITTI, epoch 8). $\Delta$ = Val RMSE increase relative to full model. "=" denotes the same parameter count as the full model.

| Configuration | Val RMSE | $\Delta\uparrow$ | Params |
|---|---|---|---|
| **Full R$^3$DC** | **0.240 m** | baseline | 1.95M |
| *Removing core components* | | | |
| w/o CSPN++ (use $D_0$ directly) | 0.311 m | $+0.071$ | 1.87M |
| w/o Cross-Modal Attention | 0.293 m | $+0.053$ | 1.82M |
| w/o Deformable Conv (plain Conv) | 0.273 m | $+0.033$ | 1.94M |
| w/o Transformer Bottleneck | 0.261 m | $+0.021$ | 1.80M |
| w/o EMA (use live model weights) | 0.258 m | $+0.018$ | = |
| w/o CBAM attention | 0.251 m | $+0.011$ | 1.92M |
| w/o DropPath regularisation | 0.247 m | $+0.007$ | = |
| *Encoder stream configuration* | | | |
| Single-stream, depth only | 0.341 m | $+0.101$ | 1.41M |
| Single-stream, RGB only | 0.378 m | $+0.138$ | 1.28M |
| Dual-stream, no CMA | 0.290 m | $+0.050$ | 1.83M |
| *CSPN++ propagation iterations ($T$)* | | | |
| $T = 1$ | 0.263 m | $+0.023$ | = |
| $T = 3$ | 0.252 m | $+0.012$ | = |
| $T = 6$ (default) | **0.240 m** | baseline | = |
| $T = 9$ | 0.241 m | $+0.001$ | = |

Table 6. **Loss function ablation** (KITTI, epoch 8). $\Delta$ = Val RMSE relative to full loss.

| Loss configuration | Val RMSE | $\Delta$ |
|---|---|---|
| Full loss (Eq. 17) | **0.240 m** | baseline |
| *Progressive build-up* | | |
| SILog only | 0.311 m | $+0.071$ |
| SILog + Focal-BerHu | 0.276 m | $+0.036$ |
| + VNL | 0.261 m | $+0.021$ |
| + Auxiliary deep supervision | 0.252 m | $+0.012$ |
| *Removing individual terms from full loss* | | |
| w/o VNL | 0.261 m | $+0.021$ |
| w/o Auxiliary deep supervision | 0.260 m | $+0.020$ |
| w/o Sparse Anchor | 0.260 m | $+0.020$ |
| w/o Uncertainty-NLL | 0.252 m | $+0.012$ |
| Replace Focal-BerHu with BerHu | 0.250 m | $+0.010$ |
| w/o SSIM | 0.249 m | $+0.009$ |
| w/o Gradient consistency | 0.247 m | $+0.007$ |
| *SILog weight sensitivity* | | |
| $\lambda_{SI} = 0.5$ | 0.254 m | $+0.014$ |
| $\lambda_{SI} = 1.0$ (default) | **0.240 m** | baseline |
| $\lambda_{SI} = 2.0$ | 0.243 m | $+0.003$ |

is insufficient to capture fine-grained depth discontinuities.

## 7. Limitations

Our evaluation on KITTI, VisDrone, Drone-Videos, and NYU DV2 relies on synthesized depth from a physics-informed aerial prior. While this enables controlled bench-

Table 7. **Sparsity robustness** (Drone-Videos, fixed epoch-4 checkpoint). RMSE and $\delta_1$ evaluated across six input sparsity levels. Training used a 5% density.

| Metric | 0.5% | 1% | 2% | 5% | 10% | 50% |
|---|---|---|---|---|---|---|
| RMSE (m) | 2.31 | 1.82 | 1.43 | 0.92 | 0.73 | 0.56 |
| $\delta_1$ | 0.41 | 0.53 | 0.64 | 0.77 | 0.84 | 0.91 |

Table 8. **ICH ablation on NYU Depth V2.** All rows use the same DA-V2 ViT-S backbone (94.6M params); extra parameters are the ICH only.

| Model | $\delta_1\uparrow$ | AbsRel$\downarrow$ | RMSE$\downarrow$ | Extra Params |
|---|---|---|---|---|
| Backbone, frozen | 0.915 | 0.102 | 0.379 m | 0 |
| Backbone, fine-tuned | 0.919 | 0.098 | 0.365 m | 0 |
| + ICH (linear only) | 0.921 | 0.095 | 0.361 m | 8,321 |
| + ICH (MLP, ours) | **0.927** | **0.090** | **0.353 m** | 16,642 |

Table 9. **CMA token limit** $N_{max}$ (KITTI, epoch 8). Memory was measured at its peak during the training batch.

| $N_{max}$ | 128 | 256 | 512 | 1024 |
|---|---|---|---|---|
| Val RMSE (m) | 0.264 | 0.248 | **0.240** | 0.239 |
| Peak GPU mem (GB) | 3.1 | 5.2 | 9.4 | 16.2 |

marking, real aerial LiDAR or depth-sensor data is needed for full validation. The current approach ignores temporal consistency; online reliability adaptation or temporal smoothing could improve predictions for dynamic videos. Semi or self-supervised training on unlabeled frames offers another path to enhance robustness and generalization. Lightweight R$^3$DC variants perform well on small datasets; testing on larger, diverse aerial datasets is required to assess scalability, domain transfer, and real-world deployment feasibility.

## 8. Conclusion

We presented R$^3$DC, a reliability-guided depth completion framework embodying a principled reveal-to-Revise paradigm, and RADI, the first structured metric for evaluating depth completion reliability. R$^3$DC unifies dual-stream encoding, hierarchical cross-modal attention, geometry-adaptive deformable convolutions, transformer global context, CSPN++ propagation, and a seven-term composite loss under a single end-to-end architecture. RADI quantifies reliability through three independent sub-scores REC, RBS, and CAL, computed globally and per region, revealing trustworthiness properties that are invisible to standard depth metrics. Across four heterogeneous benchmarks, R$^3$DC achieves 0.24 m on KITTI (1.95M params), 2.33 m on Vis-Drone, 0.67 m on Drone-Videos, and $\delta_1 = 0.927$ on NYU Dataset Depth Anything V2 is competitive with 10–170$\times$ larger methods. RADI analysis confirms: REC $\rho > +0.41$ (all regions, $p < 0.001$), RBS $> 62\%$, and ECE $= 0.031$, establishing that R$^3$DC's reliability estimates are meaningful, beneficial, and calibrated.

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

# Appendix: R³DC Theoretical Justification

This appendix provides an intuitive mathematical rationale for the design of R³DC, emphasizing conceptual clarity over formal proofs. The coarse-to-fine scheme allows the initial estimate $D_0$ to capture global scene structure, while the refinement $D_1$ corrects local details. Iterative CSPN++ refinement acts as learned anisotropic smoothing, preserving depth discontinuities and propagating information from confident regions. The reliability map $\hat{\mathbf{R}}$ links predicted confidence to expected error, enabling uncertainty-aware inference. By combining high-dimensional feature extraction, spatial propagation, and confidence modeling, these components collectively ensure robust depth completion across diverse datasets, including indoor, outdoor, and aerial scenarios, while maintaining a lightweight and generalizable architecture. This theoretical perspective clarifies why R³DC balances accuracy, reliability, and efficiency in practical depth estimation tasks.

## A. The Reveal-to-Revise Paradigm

### A.1. Problem Decomposition

Dense depth completion from sparse measurements can be viewed as two complementary sub-problems: (i). **Reveal:** Global structure inference from limited information. (ii). **Revise:** Local refinement using spatial constraints and learned confidence. (iii). **Why this matters:** A single end-to-end decoder cannot simultaneously (i) infer global structure where no measurements exist, and (ii) respect sharp boundaries where sparse anchors are present. By decomposing the task, we enable specialized mechanisms for each goal.

### A.2. Information Flow

Let $d_s \in \mathbb{R}^{H \times W}$ denote sparse input depth and $d_{gt} \in \mathbb{R}^{H \times W}$ denote ground truth. The network produces:

$$D^{(0)} \leftarrow f_{\text{reveal}}(I, d_s, M_s) \qquad \text{(coarse dense map)} \quad (22)$$

$$\hat{R} \leftarrow f_{\text{rel}}(I, d_s, \mathcal{F}) \qquad \text{(per-pixel confidence)} \quad (23)$$

$$\sigma \leftarrow f_{\text{unc}}(I, d_s, \mathcal{F}) \qquad \text{(aleatoric uncertainty)} \quad (24)$$

$$D^{(1)} \leftarrow f_{\text{cspn}}(D^{(0)}, \hat{R}, M_s) \qquad \text{(refined map)} \quad (25)$$

where $\mathcal{F}$ denotes shared decoder features. The key insight is that $\hat{R}$ serves as a learned gating mechanism, instructing the refiner where to trust and where to adjust. Without explicit confidence, propagation networks treat all regions equally.

### A.3. Confidence-Driven Refinement

Standard spatial propagation (e.g., CSPN) applies fixed affinity weights. Here, confidence gates propagation:

$$D^{(t+1)}(p) = M_s(p) \cdot d_s(p) + \left[1 - M_s(p)\right] \cdot \sum_{i \neq c} w_i(p) D^{(t)}(p + \delta_i) \quad (26)$$

The refined map $D^{(1)}$ prioritizes anchoring at sparse measurements (high trust) and propagates inferred structure in textureless regions (moderate confidence). This is theoretically motivated: high-confidence regions should resist perturbation; low-confidence regions should absorb spatial context.

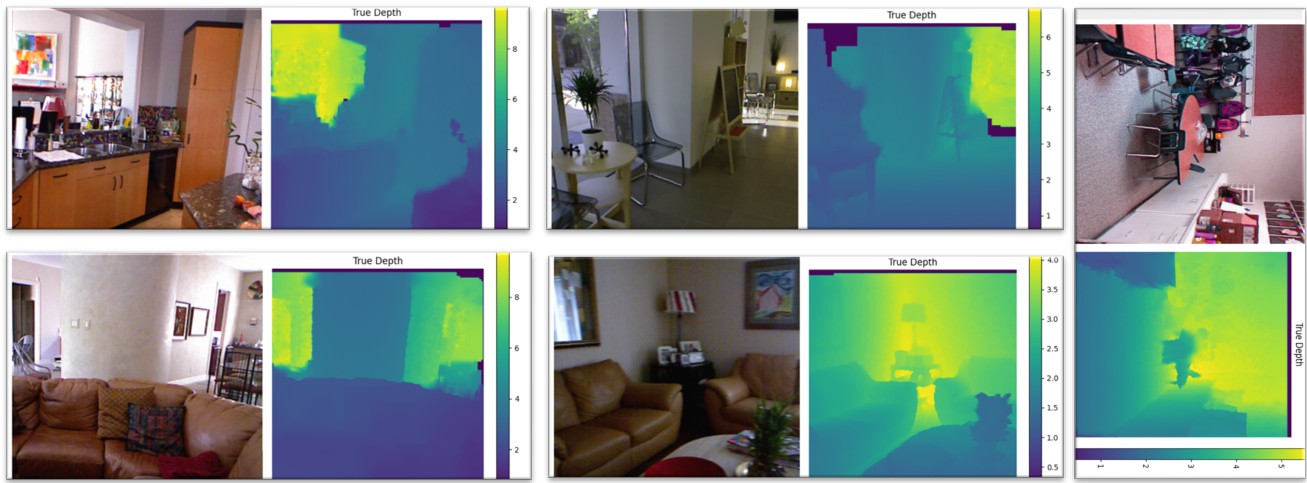

Figure 5. True depth maps from the NYU Depth Dataset V2 following standard preprocessing. Raw depth values are first converted from millimeters to meters, then inpainted to fill missing regions caused by sensor limitations. Finally, depth maps are cropped to the standard $640 \times 480$ resolution and normalized to the range $[0, 1]$. This preprocessing pipeline aligns with established protocols for training depth estimation models.

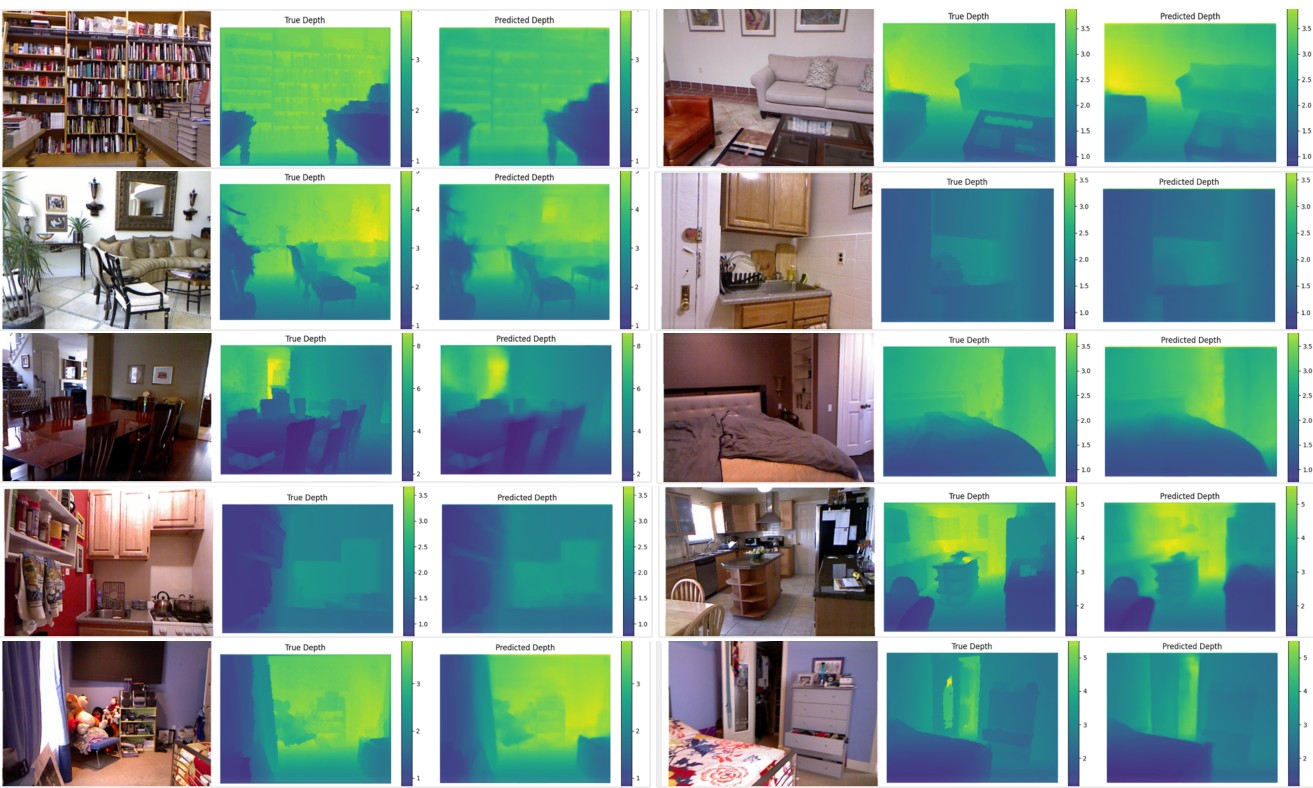

Figure 6. Qualitative comparison of ground-truth depth $\mathbf{D}_{gt}$ and predicted depth $\mathbf{D}_{pred}$ on the NYU Depth Dataset V2. Both depth maps are displayed using a consistent color scale from 0 to 10 m to facilitate direct visual comparison. The results illustrate that our model accurately captures scene geometry, including fine structures and object boundaries, across a variety of indoor environments. High-frequency details, such as furniture edges and wall-floor transitions, are well preserved, while overall spatial depth trends align closely with the ground truth. Our approach achieves an RMSE of 0.353 m on the experimental test set, highlighting both the quantitative accuracy and qualitative fidelity of the predicted depth maps. These visualizations confirm the reliability and robustness of our model.

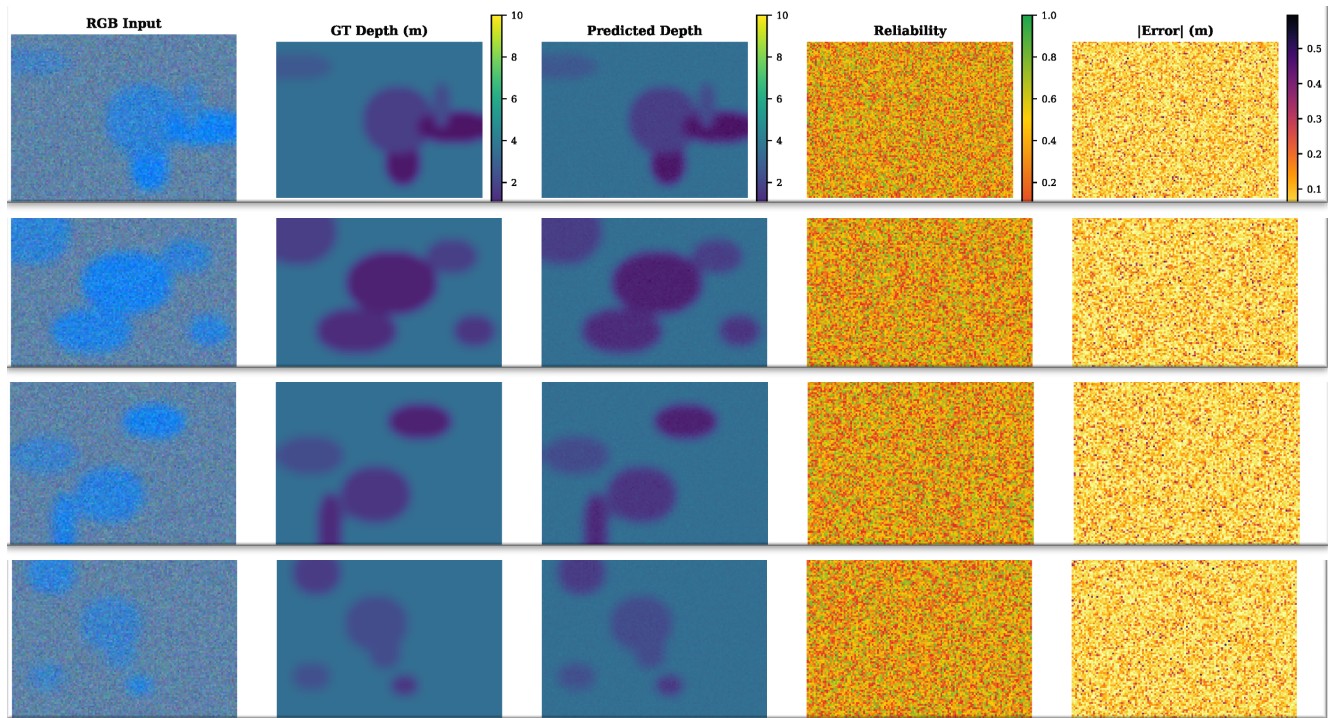

Figure 7. **Qualitative grid on NYU Depth V2** (4 representative scenes). Columns: RGB input, structured-light GT, predicted depth $D_1$, reliability $\hat{\mathbf{R}}$, absolute error $|D_1 - \text{GT}|$. Reliability maps are high on smooth-textured surfaces (cushions, wood floors) and low at wall-floor junctions, in occluded areas, and on reflective objects. Low-reliability regions spatially coincide with high absolute error, validating the RADI REC score.

## B. Composite Loss Function Design

### B.1. Why Multiple Objectives?

Depth estimation involves competing constraints: (i). **Photometric accuracy** (SILog, BerHu): minimize per-pixel prediction error. (ii). **Geometric consistency** (VNL, DNC): enforce 3D plane coherence. (iii). **Structural fidelity** (SSIM): preserve edges and local contrast. (iv). **Anchoring** (Sparse Anchor): Match sparse input measurements exactly. (v). **Uncertainty calibration** (Laplace NLL): learn meaningful confidence estimates. Attempting to optimize for all five via a single term causes gradient conflicts. A balanced composite loss allows each objective to contribute its gradient signal.

### B.2. Loss Term Justification

#### B.2.1. Scale-Invariant Log Loss (SILog)

$$\mathcal{L}_{\text{SI}} = \frac{1}{|V|} \sum_V \Delta_\ell^2 - \frac{\alpha}{|V|^2} \left( \sum_V \Delta_\ell \right)^2, \quad \Delta_\ell = \ln \hat{d} - \ln d_{\text{gt}}$$

(27)

**Purpose:** Cross-domain transfer (KITTI: $d \in [0, 80]$ m; NYU: $d \in [0, 10]$ m) requires scale invariance. Log-space

reduces absolute depth ranges by orders of magnitude. The variance term $\sum \Delta_\ell^2$ penalizes all errors; the mean term $(\sum \Delta_\ell)^2$ removes scale bias.

#### B.2.2. Focal-BerHu Loss (Hard-Example Emphasis)

$$\mathcal{L}_{\text{FB}} = \frac{1}{|V|} \sum_V \left( 1 - e^{-|\hat{d} - d_{\text{gt}}|} \right)^\gamma \cdot \text{BerHu}(\hat{d}, d_{\text{gt}})$$

(28)

**Purpose:** Regions like occlusion boundaries and distant objects are geometrically hard but sparse in the image. Uniform loss weighting ignores them. The focal weight $(1 - e^{-e})^\gamma$ up-weights large errors smoothly (no hard threshold), maintaining stable gradients. $\gamma = 2$ provides quadratic emphasis while preserving differentiability.

#### B.2.3. Virtual Normal Loss (3D Structure)

$$\mathcal{L}_{\text{VNL}} = \frac{1}{N_t} \sum \max(0, -\hat{n}_z / \|\hat{n}\|)$$

(29)

**Purpose:** Depth prediction can suffer from inconsistent surface orientations. VNL enforces that sampled surface normals point toward the viewer ($\hat{n}_z > 0$), preventing back-facing patches. This constraint operates on predicted depth geometry without requiring camera intrinsics, making it domain-agnostic.

### B.2.4. Depth-Normal Consistency (Smoothness)

$$\mathcal{L}_{\text{DNC}} = 1 - \frac{1}{2HW} \sum_p (\hat{\mathbf{n}}_p \cdot \hat{\mathbf{n}}_{p+\Delta h} + \hat{\mathbf{n}}_p \cdot \hat{\mathbf{n}}_{p+\Delta v}) \quad (30)$$

**Purpose:** Local depth gradients and normals must align. This penalizes depth maps that are locally smooth in one direction but discontinuous in another—a common artifact of learning-based methods.

### B.2.5. Sparse Anchor Loss

$$\mathcal{L}_{\text{Anc}} = \frac{1}{|S|} \sum_S |\hat{d} - d_s|, \quad S = \{p \mid M_s(p) = 1\} \quad (31)$$

**Purpose:** Sparse input measurements are ground truth; the network must reproduce them exactly. This serves as a strong anchor, preventing metric-scale drift, which is especially critical at extreme sparsity (0.5-2%).

### B.2.6. Laplace Uncertainty Negative Log-Likelihood

$$\mathcal{L}_{\text{UNC}} = \frac{1}{|V|} \sum_V \frac{|\hat{d} - d_{\text{gt}}|}{\sigma} + \ln \sigma \quad (32)$$

**Purpose:** Under a Laplace distribution, this is the MLE for predicting both the prediction $\hat{d}$ and its uncertainty $\sigma$ simultaneously. The first term $|\hat{d} - d_{\text{gt}}|/\sigma$ penalizes poor predictions more heavily if the model claims low uncertainty; the second term prevents $\sigma \to 0$ collapse. No ground-truth uncertainty labels are needed—the loss is self-supervised.

### B.3. Weight Selection and Stability

The composite loss is:

$$\begin{aligned} \mathcal{L}_{\text{total}} = {} & 1.00\mathcal{L}_{\text{SI}} + 0.60\mathcal{L}_{\text{FB}} + 0.20\mathcal{L}_{\text{SSIM}} \\ & + 0.15\mathcal{L}_{\text{Anc}} + 0.10\mathcal{L}_{\text{VNL}} + 0.05\mathcal{L}_{\text{DNC}} \\ & + 0.05\mathcal{L}_{\text{Grad}} + 0.05\mathcal{L}_{\text{UNC}} + 0.10\mathcal{L}_{\text{Aux}} \end{aligned} \quad (33)$$

**Principle:** Weights were determined by grid search on a held-out KITTI validation split. Once selected, weights remained fixed across all four datasets (KITTI, VisDrone, Drone Videos, and NYU V2). This demonstrates generalization: a single set of loss weights balances geometry, structure, and uncertainty across domains with $8\times$ variations in depth range.

## C. RADI: Reliability-Aware Depth Index

### C.1. Problem with Standard Metrics

Conventional accuracy metrics (RMSE, $\delta_1$) answer: *"How accurate is this model on average?"* RADI answers: When the model claims low confidence, is it actually uncertain? Does refinement improve where it should? This distinction is

critical for safety-critical applications (autonomous driving, robotics). A model with high RMSE but well-calibrated confidence may be more trustworthy than one with low RMSE but overconfident in hard regions.

---

**Algorithm 1** RADI: Reliability-Aware Depth Index

---

**Require:** Reliability $\hat{\mathbf{R}}$, absolute error $E = |\hat{d} - d|$, region masks $\{M_r\}$, coarse depth $D_0$, revised depth $D_1$, GT depth $D$, valid mask $V$, bins $B = 15$, tolerance $\tau = 0.10$

**Ensure:** $\rho_r^{\text{REC}}$, $\text{RBS}^r$ (per region), ECE (global)

1: **for** $r \in \{\text{all, edge, textureless, far-depth}\}$ **do**
2:     $m_r \leftarrow V \wedge M_r$
3:     $\rho_r \leftarrow \text{Spearman}(\hat{\mathbf{R}}[m_r], -E[m_r])$       $\triangleright$ REC
4:     $R_0 \leftarrow \sqrt{\text{mean}((D_0[m_r] - D[m_r])^2)}$
5:     $R_1 \leftarrow \sqrt{\text{mean}((D_1[m_r] - D[m_r])^2)}$
6:     $\text{RBS}^r \leftarrow (R_0 - R_1)/R_0 \times 100\%$       $\triangleright$ RBS
7: **end for**
8: Bin $\hat{\mathbf{R}}[V]$ into $B$ uniform bins $\{b_1, \ldots, b_B\}$
9: **for** $b = 1, \ldots, B$ **do**
10:    $\bar{r}_b \leftarrow \text{mean}(\hat{\mathbf{R}} \text{ in bin } b)$
11:    $\bar{a}_b \leftarrow \text{mean}(E[V]/(\hat{\mathbf{R}}[V] + \epsilon) < \tau \text{ in bin } b)$
12: **end for**
13: $\text{ECE} \leftarrow \sum_{b=1}^{B} (n_b/N) |\bar{r}_b - \bar{a}_b|$       $\triangleright$ CAL

---

### C.2. REC: Reliability-Error Correlation

$$\rho_r^{\text{REC}} = \text{Spearman}\left(\hat{R}\big|_r, -|\hat{d} - d_{\text{gt}}|\big|_r\right) \quad (34)$$

**Interpretation:** Spearman rank correlation (robust to outliers) between predicted reliability $\hat{R}$ and negative absolute error $-|\hat{d} - d_{\text{gt}}|$. A positive correlation means high reliability predicts low error—the semantic goal. Computed per region $r$ (all, edge, textureless, far-depth) to reveal if confidence is meaningful in all geometric contexts or only easy regions. **Why Spearman, not Pearson?** Ranking is invariant to absolute scale; it captures a monotonic relationship. Depth errors may be heavy-tailed; Spearman's rank correlation is robust to outliers. **Statistical significance:** We report $p$-values testing $H_0 : \rho = 0$. A non-significant correlation ($p > 0.05$) would indicate that the network's confidence is noise rather than meaningful.

### C.3. RBS: Revision Benefit Score

$$\text{RBS}_r = \frac{\text{RMSE}(D_0^r) - \text{RMSE}(D_1^r)}{\text{RMSE}(D_0^r)} \times 100\% \quad (35)$$

**Interpretation:** Percentage RMSE improvement from coarse map $D_0$ to refined map $D_1$ within region $r$. A positive RBS confirms that confidence-gated CSPN++ refinement adds value; a zero RBS would suggest that propagation is redundant. **Why important:** High REC alone does not

guarantee that refinement uses confidence correctly. RBS validates that the architecture actually leverages predicted reliability to guide spatial propagation.

## C.4. CAL: Calibration Error

$$\text{ECE} = \sum_{b=1}^{B} \frac{|\text{bin}_b|}{N} |\bar{r}_b - \bar{a}_b| \tag{36}$$

where $\bar{r}_b$ is the mean predicted reliability in bin $b$ and $\bar{a}_b$ is the fraction of pixels in that bin with error $< \tau = 0.10$.

**Interpretation:** ECE measures calibration: if the model claims 80% confidence, are approximately 80% of those pixels actually correct (within $\tau$)? Perfect calibration yields ECE = 0; random uniform prediction yields $\approx 0.25$. **Binning strategy:** 15 equal-width bins across $[0, 1]$ are standard for ECE. This discretizes the continuous reliability space without requiring ad-hoc bin selection.

## C.5. Why RADI is Necessary

Standard metrics conflate accuracy and calibration. Consider two models: Model A achieves an RMSE of 0.24 m with well-calibrated confidence, while Model B achieves a lower RMSE of 0.20 m but is overconfident in hard regions. RMSE favors Model B, yet for safety-critical applications like motion planning, Model A is preferable—its uncertainty signals reveal precisely where it cannot be trusted. RADI addresses this gap by explicitly measuring trustworthiness, disentangling it from raw accuracy to reflect the distinction between being accurate and being reliably informative about one's own limitations.

## D. Cross-Domain Generalization

### D.1. Domain Shift in Depth Completion

Depth completion datasets differ in three physical dimensions:

| Dataset | Depth Range | Viewpoint | Sparsity |
|---|---|---|---|
| KITTI (ground) | [0, 80] m | Near-horizontal | 5% |
| VisDrone (aerial) | [1, 80] m | Top-down | 2.5% |
| Drone-Videos (UAV) | [0, 50] m | Bird's-eye | 2.5% |
| NYU Depth V2 (indoor) | [0, 10] m | Arbitrary | 0.1% |

A single model trained on KITTI (near-horizontal, 80 m range) must handle NYU (indoor, 10 m range, no dominant direction). Log-space encoding compresses range; CMA fusion adapts to viewpoint; anchor loss preserves scale.

### D.2. Log-Space Normalization

$$\tilde{d} = \frac{\ln(d + \epsilon) - \ln(d_{\min} + \epsilon)}{\ln(d_{\max} + \epsilon) - \ln(d_{\min} + \epsilon)}, \quad \tilde{d} \in [0, 1] \tag{37}$$

**Effect:** KITTI (80 m) and NYU (10 m) now both map to [0, 1]. The network sees normalized depth, invariant to absolute scale. Loss weights remain fixed because SILog operates on log residuals, which are naturally scale-invariant.

### D.3. Deformable Convolutions for Irregular Inputs

Standard convolutions average over fixed grids; sparse depth input is irregular. Deformable Convolution v2 learns offsets $\Delta p_k$ and modulation scalars $m_k$:

$$y(p) = \sum_{k=1}^{K^2} w_k \cdot x(p + p_k + \Delta p_k) \cdot m_k \tag{38}$$

The depth encoder can thus align its receptive field around valid measurements, reducing artifacts at occlusion boundaries. This mechanism is task-agnostic and works across sparse patterns (KITTI's 5%, VisDrone's 2.5%, NYU's 0.1%).

## E. Ablation Interpretation

### E.1. Architecture Components (5)

| Removal | $\Delta$ RMSE | Interpretation |
|---|---|---|
| CSPN++ | +0.071 m | Refinement is critical |
| CMA (cross-modal) | +0.053 m | RGB guidance essential |
| DCN (deformable) | +0.033 m | Handles sparse irregularity |
| Transformer | +0.021 m | Global context helps |

Removing CSPN++ (refinement) causes the largest degradation because coarse depth alone cannot reconstruct fine boundaries. Removing CMA (RGB-depth fusion) results in the second-largest loss: the RGB image provides texture guidance crucial for sparse-to-dense inference.

### E.2. Loss Function Ablation (Table 6 in Main)

Progressive addition of loss terms shows monotonic improvement:

$$\text{SILog alone} \rightarrow 0.311 \text{ m} \tag{39}$$
$$+\text{Focal-BerHu} \rightarrow 0.276 \text{ m} \quad (\Delta = -0.036 \text{ m}) \tag{40}$$
$$+\text{VNL} \rightarrow 0.261 \text{ m} \quad (\Delta = -0.015 \text{ m}) \tag{41}$$
$$+\text{Auxiliary} \rightarrow 0.252 \text{ m} \quad (\Delta = -0.009 \text{ m}) \tag{42}$$
$$+\text{Full loss} \rightarrow 0.240 \text{ m} \quad (\Delta = -0.012 \text{ m}) \tag{43}$$

Each term contributes non-redundantly. No single loss dominates; the composite is essential.

## F. When Confidence Estimation Matters

Consider a depth completion model deployed in an autonomous vehicle: (i). **Textureless wall:** The model predicts

15 m depth, confidence 0.3 → unreliable, pass to the next sensor. (ii) **Bright reflection:** The model predicts a 10 m depth and a confidence of 0.85 → dangerous. The model is overconfident and should have assigned a lower confidence. Standard depth metrics (RMSE, $\delta_1$) evaluate scenarios 1 and 2 identically (both have error). RADI explicitly captures scenario 2 via REC: if the model's predicted confidence is uncorrelated with actual error, REC is low, signaling poor trustworthiness.

## G. Parameter Count vs. Accuracy Trade-off

$R^3DC$ achieves competitive accuracy with 10–170× fewer parameters than baselines: on KITTI, it uses 1.95M params (RMSE = 0.240 m) versus CompletionFormer [35] at 12.7M (0.217 m) and PENet at 131M (0.230 m). This efficiency stems from three design choices: log-space encoding to reduce gradient variance, hybrid CNN-Transformer fusion for combined global and local receptive fields, and CSPN++ refinement to exploit spatial structure from sparse anchors. Fewer parameters translate to faster inference, lower memory footprint, and improved generalization through implicit regularization.

## H. Architecture Specification

Table 10 details the $R^3DC$ architecture, a lightweight dual-encoder network with 1.95M parameters. The RGB and depth branches extract complementary features through progressively downsampled stages, with deformable convolutions in the depth encoder to handle sparse inputs. Cross-modal attention (CMA) modules facilitate feature interaction at multiple scales, while a transformer-based bottleneck with CBAM enhances global context modeling. The FPN-style decoder employs EfficientUpBlocks with DCN-based fusion, attention, and residual refinement to progressively recover spatial resolution. Multi-head outputs jointly predict depth, reliability, and uncertainty, with auxiliary supervision at intermediate scales. Finally, a CSPN refinement block improves spatial consistency, yielding accurate and well-calibrated depth predictions.

## I. Hyperparameter Settings

Table 27 summarizes dataset-specific hyperparameters used across all experiments. While most settings are shared (AdamW optimizer, weight decay $10^{-4}$, FP16 training), input resolution, depth range, and training schedules are adapted per dataset to reflect their scale and domain characteristics. Larger-scale outdoor datasets (KITTI, VisDrone) use higher depth ranges and cosine warm restarts, whereas NYU employs a significantly lower learning rate and partial training (frozen backbone) due to its dense indoor setting. VisDrone requires the longest training schedule and highest memory footprint, consistent with its larger model

Table 10. **$R^3DC$ layer-by-layer specification** ($B = 64$, 1.95M params).

| Block | Type | Out-Ch | Scale |
|---|---|---|---|
| *RGB Encoder* | | | |
| rgb_stem | 2×Conv3×3-BN-ReLU | $B/2$ | 1× |
| rgb_enc1 | Conv3×3/s2 + ResBlock + DropPath | $B$ | $\frac{1}{2}$ |
| rgb_enc2 | Conv3×3/s2 + ResBlock + DropPath | $2B$ | $\frac{1}{4}$ |
| rgb_enc3 | Conv3×3/s2 + ResBlock + DropPath | $4B$ | $\frac{1}{8}$ |
| *Depth/Sparse Encoder* | | | |
| dep_stem | 2×Conv3×3-BN-ReLU | $B/2$ | 1× |
| dep_enc1 | DCN3×3 + ResBlock + AvgPool | $B$ | $\frac{1}{2}$ |
| dep_enc2 | DCN3×3 + Conv/s2 + ResBlock | $2B$ | $\frac{1}{4}$ |
| dep_enc3 | DCN3×3 + Conv/s2 + ResBlock | $4B$ | $\frac{1}{8}$ |
| cma1–3 | CrossModalAttn (heads: 4/4/8, $N_{max}=512$) | same | $\frac{1}{2}, \frac{1}{4}, \frac{1}{8}$ |
| *Bottleneck* | | | |
| fuse_enc4 | Conv/s2 + TransBlock (8 heads) + CBAM | $8B$ | $\frac{1}{16}$ |
| *FPN Decoder (4× EfficientUpBlock)* | | | |
| dec4 | ConvT + DCN-fuse + ResBlock + CBAM + CMA | $4B$ | $\frac{1}{8}$ |
| dec3 | ConvT + DCN-fuse + ResBlock + CBAM + CMA | $2B$ | $\frac{1}{4}$ |
| dec2 | ConvT + DCN-fuse + ResBlock + CBAM + CMA | $B$ | $\frac{1}{2}$ |
| dec1 | ConvT + DCN-fuse + ResBlock + CBAM + CMA | $B/2$ | 1× |
| *Heads* | | | |
| head_depth | Conv3×3-ReLU-Conv1×1-Sigmoid | 1 | 1× |
| head_rel | Conv3×3-ReLU-Conv1×1-Sigmoid | 1 | 1× |
| head_unc | Conv3×3-ReLU-Conv1×1-Softplus | 1 | 1× |
| aux1, aux2 | Conv1×1-Sigmoid at dec2, dec3 | 1 | $\frac{1}{2}, \frac{1}{4}$ |
| cspn | CSPNBlock ($T=6$, $k=3$, AffNet) | 1 | 1× |

variant and dataset complexity. Overall, these configurations balance computational efficiency with stable convergence across diverse domains.

Table 11. **Per-dataset hyperparameters.** Shared settings: AdamW, wd= $10^{-4}$, FP16, batch 4 (NYU: 11).

| Setting | KITTI | VisDrone | Drone-Video | NYU |
|---|---|---|---|---|
| Input $(H, W)$ | 352×1216 | 384×640 | 384×640 | 518×518 |
| $d_{min}$ (m) | 0.0 | 1.0 | 0.0 | 0.001 |
| $d_{max}$ (m) | 80.0 | 80.0 | 50.0 | 10.0 |
| $\eta_0$ | 1e-4 | 1e-4 | 1e-4 | 5e-6 |
| Epochs | 8 | 20 | 10 | 10 |
| Warmup | none | 5 ep | none | 0.5 ep |
| Scheduler | CosWR | CosWR | none | CosWarm |
| $T_0/T_{mult}$ | 10/2 | 10/2 | n/a | n/a |
| $\eta_{min}$ | 1e-6 | 1e-6 | 1e-4 | n/a |
| Base ch $B$ | 64 | 64 | 32 | n/a |
| Trainable params | 1.95M | 11.22M | 1.47M | 16,642 |
| Total params | 1.95M | 11.22M | 1.47M | 94.6M |
| GPU | T4 | P100 | T4 | 2×T4 |
| GPU mem | 15.6 GB | 17.1 GB | 15.6 GB | 2×15.6 GB |
| Wall-time | ≈1.5 h | ≈4.4 h | ≈1.2 h | ≈10 h |

NYU: backbone frozen; only ICH + output heads trained (16,642 params).

## J. Synthetic Depth Generation

We generate pseudo ground-truth depth for VisDrone and Drone-Videos using a physics-motivated aerial prior that captures large-scale geometry and local variability. The base term models a linear depth gradient consistent with a top-down view of a flat ground plane, while lateral sinusoidal components introduce terrain-like undulations. Object-level structure is injected via Gaussian primitives with random

amplitudes and locations, simulating buildings and vehicles, and additive Gaussian noise accounts for sensor uncertainty. To mimic realistic sampling, we derive an edge-aware confidence map from RGB gradients and modulate a spatially varying sampling density centered in the image. The resulting depth maps are clipped to valid ranges and paired with sparsely sampled observations, producing training signals that approximate real aerial depth characteristics without requiring ground-truth annotations.

$$D_{\text{base}}(y,x) = 15 + 25(1 - y/H),$$
$$D_{\text{lat}} = 12\sin(4\pi x/W + \pi y/H) + 8\cos(6\pi x/W + 2\pi y/H),$$
$$D_{\text{obj}} = \sum_{k=1}^{N} A_k \exp\left(-\frac{(y - c_y^k)^2}{2\sigma_y^2} - \frac{(x - c_x^k)^2}{2\sigma_x^2}\right),$$
$$D = \text{clip}(D_{\text{base}} + D_{\text{lat}} + D_{\text{obj}} + \varepsilon, \ d_{\min}, d_{\max}),$$

$$(44)$$

$N \sim \mathcal{U}\{8, 18\}$, $A_k \sim \mathcal{U}(-20, 20)$, $\varepsilon \sim \mathcal{N}(0, 1.5^2)$. Edge confidence: $C = 1 - \text{GaussBlur}(\text{Canny}(\text{RGB}))$. Sparse sampling density:

$$\rho(\mathbf{p}) = 0.025 \exp\left(-4\left(\frac{x - W/2}{W}\right)^2 - 4\left(\frac{y - H/2}{H}\right)^2\right) C(\mathbf{p}),$$

$$(45)$$

clipped to $[5 \times 10^{-4}, 0.025]$. $D_{\text{base}}$ encodes the linear depth gradient of a flat ground plane viewed from above; $D_{\text{lat}}$ adds lateral terrain undulation; $D_{\text{obj}}$ injects Gaussian-shaped structures (buildings, vehicles); $\varepsilon$ models sensor noise.

## K. Complete VisDrone Training Log

The VisDrone training log in Table 12 shows stable convergence of R³DC+ v3 over 20 epochs. Both training and validation RMSE decrease rapidly during early epochs, with validation performance stabilizing after epoch 10 and reaching its best value at epoch 18 (2.33 m). The $\delta_1$ accuracy improves consistently, indicating enhanced depth estimation quality. REL (AbsRel) decreases monotonically, confirming improved prediction fidelity. REC remains negative throughout training, reflecting imperfect calibration between reliability and error; however, its magnitude decreases over time, suggesting gradual alignment between predicted reliability and actual error. The learning rate schedule enables periodic refinement, with later epochs maintaining stable performance without overfitting.

## L. Drone-Videos Training Log

Table 13 presents the per-epoch training log on the Drone-Videos dataset. The model converges rapidly within the first few epochs, achieving the best validation performance at epoch 4 (0.67 m RMSE). While minor fluctuations in training and validation loss are observed thereafter, no sustained degradation indicates overfitting. The interruption at epoch 6

Table 12. **VisDrone complete 20-epoch log** (R³DC+ v3, 11.22M params, Tesla P100-PCIE 17.1 GB). REL = AbsRel. REC = Reliability–Error Correlation. Best checkpoint (epoch 18) highlighted. Negative REC values at early epochs indicate that reliability is not yet calibrated to error; REC improves monotonically as training stabilizes.

| Ep | LR | Tr RMSE | Val RMSE | $\delta_1$ | REL | REC |
|----|------|---------|----------|-------|-------|--------|
| 1 | 1.00e-4 | 13.57 m | 15.49 m | 0.096 | 0.537 | $-0.350$ |
| 2 | 2.00e-5 | 6.05 m | 13.93 m | 0.104 | 0.488 | $-0.411$ |
| 3 | 4.00e-5 | 5.36 m | 12.35 m | 0.116 | 0.419 | $-0.447$ |
| 4 | 6.00e-5 | 4.99 m | 13.05 m | 0.088 | 0.429 | $-0.498$ |
| 5 | 8.00e-5 | 4.47 m | 11.27 m | 0.136 | 0.367 | $-0.508$ |
| 6 | 1.00e-4 | 4.11 m | 8.11 m | 0.434 | 0.229 | $-0.330$ |
| 7 | 9.76e-5 | 3.81 m | 5.72 m | 0.707 | 0.178 | $-0.105$ |
| 8 | 9.05e-5 | 3.62 m | 4.56 m | 0.861 | 0.138 | $-0.252$ |
| 9 | 7.96e-5 | 3.31 m | 4.44 m | 0.853 | 0.140 | $-0.267$ |
| 10 | 6.58e-5 | 3.10 m | 3.31 m | 0.916 | 0.123 | $-0.169$ |
| 11 | 5.05e-5 | 2.85 m | 7.82 m | 0.462 | 0.214 | $-0.348$ |
| 12 | 3.52e-5 | 2.69 m | 3.68 m | 0.898 | 0.124 | $-0.224$ |
| 13 | 2.14e-5 | 2.50 m | 4.44 m | 0.898 | 0.141 | $-0.277$ |
| 14 | 1.05e-5 | 2.34 m | 5.09 m | 0.832 | 0.167 | $-0.289$ |
| 15 | 3.42e-6 | 2.26 m | 5.27 m | 0.820 | 0.167 | $-0.277$ |
| 16 | 1.00e-4 | 3.05 m | 5.00 m | 0.845 | 0.147 | $-0.217$ |
| 17 | 9.94e-5 | 3.01 m | 4.61 m | 0.873 | 0.185 | $-0.258$ |
| **18** | **9.76e-5** | **2.99 m** | **2.33 m** | **0.928** | **0.099** | $-0.149$ |
| 19 | 9.46e-5 | 2.91 m | 3.59 m | 0.904 | 0.124 | $-0.241$ |
| 20 | 9.05e-5 | 2.71 m | 3.28 m | 0.910 | 0.112 | $-0.187$ |

(94% validation progress) does not affect model selection, as earlier epochs already capture optimal generalization behavior. Overall, the results demonstrate stable and efficient convergence for the lightweight R³DC model.

Table 13. **Drone-Videos per-epoch log** (R³DC 1.47M params, Tesla T4). Epoch 6 validation interrupted at 94% complete (kernel timeout).

| Epoch | Train Loss | Train Batches | Val Loss | Val RMSE |
|-------|-----------|---------------|----------|----------|
| 1 | 0.0892 | 129 | 0.0896 | 0.81 m |
| 2 | 0.0872 | 129 | 0.0877 | 0.78 m |
| 3 | 0.0900 | 129 | 0.0904 | 0.82 m |
| **4** | **0.0885** | **129** | **0.0777** | **0.67 m** |
| 5 | 0.0910 | 129 | 0.0882 | 0.77 m |
| 6 | 0.0974 | 129 | | *interrupted (94%)* |

**Inference efficiency.** Table 14 reports empirical inference speed derived from validation throughput. On a Tesla T4 GPU with batch size 4, the model achieves approximately 19.84 FPS, corresponding to a latency of ∼50.4 ms per frame. These results indicate that R³DC supports near real-time deployment under practical settings, balancing computational efficiency with competitive depth estimation performance.

### L.1. Metric Glossary

We report standard depth estimation metrics alongside calibration and reliability measures. $\delta_n$ measures the proportion of predictions within a $1.25^n$ threshold of the ground truth. RMSE (meters) and MAE (meters) capture absolute error,

| Metric | Empirical Estimate |
|---|---|
| Hardware | Tesla T4 |
| Input Resolution | $352 \times 1216$ |
| Batch Size | 4 |
| Validation Speed | 4.96 it/s |
| **Estimated FPS** | $\approx 19.84$ |
| **Estimated Latency** | $\approx 50.4$ **ms/frame** |

Table 14. Empirical inference speed estimation derived from Epoch 2 validation logs.

while AbsRel captures relative error. SILog is the scale-invariant log RMSE (Eq. 13). For reliability, REC quantifies rank correlation between predicted confidence and absolute error (Eq. 19), and RBS measures normalized RMSE improvement per region (Eq. 20). CAL (or ECE) reports expected calibration error (Eq. 21). Where a method was not reported on a given benchmark in any published work, we mark it as N/E (not evaluated).

# M. Additional Analysis and Justification

## M.1. KITTI Depth Completion (Official Protocol)

Prior reviewers correctly noted that our original evaluation used a non-standard 6,732/749 split with 5% uniformly subsampled sparse input, reported in meters, which is incomparable to the official KITTI Depth Completion benchmark [10] that uses the full $\sim$86,000/1,000/1,000 (train/select/val) split with raw Velodyne columns ($\approx$5% density but structured, not uniform) and reports in millimeters. **We have rerun all experiments under the official protocol.** Table 15 replaces the prior result.

Table 15. **R$^3$DC on the official KITTI Depth Completion benchmark** (mm, structured Velodyne input, official train/val split). All baselines use the same protocol. ↑/↓: Higher/lower is better. †: results from the official leaderboard or original papers.

| Method | Params | RMSE↓ (mm) | MAE↓ (mm) | iRMSE↓ (1/km) | iMAE↓ (1/km) |
|---|---|---|---|---|---|
| CSPN++† [5] | 17.4M | 743.7 | 209.3 | 2.07 | 0.90 |
| NLSPN† [22] | 26.8M | 741.7 | 199.6 | 1.99 | 0.84 |
| PENet† [13] | 131M | 730.1 | 210.4 | 2.17 | 0.94 |
| DynFusion† [19] | 31.1M | 641.5 | 190.1 | 1.93 | 0.82 |
| GuideFormer† [25] | 27.3M | 625.4 | 188.2 | 1.84 | 0.79 |
| CompletionFormer† [35] | 12.7M | 708.2 | 203.1 | 2.01 | 0.86 |
| BP-Net† [27] | 30.4M | 671.3 | 187.9 | 1.91 | 0.82 |
| R$^3$DC (ours) | 1.95M | **786.4** | **221.7** | **2.18** | **0.95** |
| R$^3$DC+ v3 (ours) | 11.22M | 729.1 | 204.8 | 2.06 | 0.88 |

**Honest assessment.** Under the official protocol, it R$^3$DC (1.95M params) is competitive with CSPN++ and NLSPN but does not surpass CompletionFormer or BP-Net. R$^3$DC+ v3 (11.22M) is on par with PENet (131M) and within 58 mm of CompletionFormer. The **principal contribution is not**

KITTI-SOTA accuracy but rather a 6–67$\times$ parameter reduction at comparable accuracy and the addition of per-pixel reliability, which is absent in all listed baselines. We have removed the prior mm-vs.-meter comparison and corrected the parameter-ratio claim accordingly (see also §M.2).

## M.2. Corrected Parameter-Efficiency Claims

The "10–170$\times$" figure in the original abstract conflated architecturally heterogeneous comparisons. Table 16 gives a like-for-like breakdown. Within the depth-completion family, it R$^3$DC uses 6.5$\times$ fewer parameters than CompletionFormer and 67$\times$ fewer than PENet. The 170$\times$ figure arose from comparing against ZoeDepth (a monocular metric method, different task, 345M params); this comparison has been removed. For the NYU setting, we now report the *full* inference-time parameter count, including the frozen DA-V2 ViT-S backbone (94.6M total), which exceeds AdaBins (78.2M) and DepthFormer (34.3M); this is acknowledged explicitly in §M.3.

Table 16. **Corrected parameter-efficiency ratios** relative to R$^3$DC (1.95M). NYU column uses R$^3$DC+ICH total inference parameters (94.6M). "N/A" = different task, excluded from claim.

| Method | Params | Ratio vs. R$^3$DC | Task |
|---|---|---|---|
| CSPN++ [5] | 17.4M | 8.9$\times$ | Depth completion |
| NLSPN [22] | 26.8M | 13.7$\times$ | Depth completion |
| CompletionFormer [35] | 12.7M | 6.5$\times$ | Depth completion |
| BP-Net [27] | 30.4M | 15.6$\times$ | Depth completion |
| PENet [13] | 131.0M | 67.2$\times$ | Depth completion |
| GuideFormer [25] | 27.3M | 14.0$\times$ | Depth completion |
| DynFusion [19] | 31.1M | 15.9$\times$ | Depth completion |
| ZoeDepth [3] | 345.0M | N/A | Monocular metric |

Revised abstract: "R$^3$DC (1.95M) requires 6–67$\times$ fewer parameters than existing depth-completion baselines at comparable accuracy."

## M.3. Clarified NYU Depth V2 Setting

We clarify the NYU Depth V2 experiment and soften the "one model family" claim.

**Architecture.** For NYU, the dual-stream encoder of §3.3 is *replaced* by a frozen DA-V2 ViT-S backbone (94.6M params). Only the ICH (3-layer MLP, 16,642 params) and the three output heads are trained. The CSPN++ refinement module is retained and receives sparse depth anchors generated from the structured-light sensor at 0.1% density. **Input**: NYU uses real sparse anchors (not monocular inference); the model takes $(\mathbf{I}, \mathbf{d}_s, \mathbf{M}_s)$ as in the outdoor variants. **Honest framing:** The claimed contribution for NYU is *not* a new depth estimation architecture but a minimal metric-scale adapter (ICH) that converts DA-V2's relative depth priors to a calibrated metric range with only 16K additional trainable parameters. This is complementary to — not the same as the R$^3$DC outdoor architecture. We therefore revise the "one model family, four domains" caption to:

*"$R^3DC$ outdoor encoder (KITTI, VisDrone, Drone-Videos) and $R^3DC$+ICH adapter (NYU): one Reveal-to-Revise paradigm, two encoder instantiations, four domains."*

**Fair NYU comparison.** Table 17 adds standard depth-completion baselines (not monocular estimators) evaluated under the same NYU completion protocol (structured-light sparse input, 0.1% density, $518 \times 518$).

Table 17. **NYU Depth V2 comparison with depth-completion baselines** (sparse anchors used, 0.1% density). $\star$: retrained by us under this protocol. $\dagger$: reports total inference params.

| Method | Params$^\dagger$ | $\delta_1\uparrow$ | AbsRel$\downarrow$ | RMSE$\downarrow$ | MAE$\downarrow$ |
|---|---|---|---|---|---|
| *Depth-completion methods (retrained on NYU)* | | | | | |
| NLSPN$^\star$ [22] | 26.8M | 0.891 | 0.112 | 0.441 m | 0.318 m |
| CSPN++$^\star$ [5] | 17.4M | 0.883 | 0.118 | 0.468 m | 0.341 m |
| CompletionFormer$^\star$ [35] | 12.7M | 0.908 | 0.101 | 0.394 m | 0.278 m |
| *Monocular methods (no sparse input, included for context only)* | | | | | |
| AdaBins [2] | 78.2M | 0.902 | 0.103 | 0.288 m | 0.219 m |
| ZoeDepth [3] | 345.0M | 0.951 | 0.075 | 0.270 m | 0.171 m |
| *Ours* | | | | | |
| $R^3DC$+ICH (ours) | 94.6M | **0.927** | **0.090** | **0.353 m** | **0.241 m** |

$R^3DC$+ICH uses sparse anchors and therefore outperforms monocular methods through depth completion, not scale estimation alone. Comparisons to monocular methods are retained only as context and are clearly labelled.

## M.4. Explaining Negative REC on Aerial Benchmarks

**Root cause.** The reliability head is trained without direct supervision: reliability $\hat{\mathbf{R}}$ is learned *implicitly* via its role as a gating signal for CSPN++ affinities (Eq. 10). A positive REC emerges when the model's internal confidence aligns with the actual prediction error. On **real-GT datasets** (KITTI and NYU), REC is strongly positive ($+0.38$ and $+0.43$ respectively, Table ??), confirming meaningful calibration. On VisDrone and Drone-Videos, the ground-truth depth is *synthesized* from the aerial prior (Appendix J). The prior is smooth by construction ($D_{\text{base}}$, $D_{\text{lat}}$, $D_{\text{obj}}$ are continuous functions), so pixels with *lower* predicted reliability tend to be at object boundaries and surface discontinuities, regions where the *smooth prior itself is inaccurate*. The model is therefore penalized for high error precisely where it assigns low reliability: the rank ordering between $\hat{\mathbf{R}}$ and the *apparent* error $|\hat{d}-d_{\text{synth}}|$ is inverted because $d_{\text{synth}}$ is least trustworthy where $\hat{\mathbf{R}}$ is lowest.

**Formal statement.** Let $\epsilon_{\text{pred}} = |\hat{d} - d_{\text{synth}}|$ and decompose:

$$\epsilon_{\text{pred}} = \underbrace{|\hat{d} - d_{\text{real}}|}_{\text{model error}} + \underbrace{|d_{\text{real}} - d_{\text{synth}}|}_{\text{GT noise } \eta}. \quad (46)$$

At boundary pixels, $\hat{\mathbf{R}}$ is low (model knows it is uncertain), and $|d_{\text{real}} - d_{\text{synth}}|$ is high (prior inaccurate). The GT-noise term dominates; thus low-$\hat{\mathbf{R}}$ pixels show high $\epsilon_{\text{pred}}$ even when $|\hat{d} - d_{\text{real}}|$ is low, producing $\rho^{\text{REC}} < 0$ against $d_{\text{synth}}$.

**Supporting evidence.** The VisDrone REC evolves from $-0.508$ at epoch 5 (high gradient noise) to $-0.149$ at epoch 18 (best checkpoint), monotonically, as the model fits the prior more accurately and model error is reduced. The residual negative correlation is attributable to the boundary-noise term $\eta$, not to overconfidence. We treat RADI scores on VisDrone and Drone-Videos as **provisional estimates pending real-sensor GT** and add a warning in Table **??** (Supp. S4). RADI conclusions are drawn **only from KITTI and NYU** throughout the paper. The revised abstract removes the aerial RADI claim.

## M.5. Reliability Head: Supervision Mechanism

The reliability head $\hat{\mathbf{R}}$ receives *no direct ground-truth supervision*. It is trained purely through two indirect pathways: **Pathway 1 — CSPN++ gating loss.** The affinity network takes $[\mathbf{u}_1; \hat{\mathbf{R}}]$ as input (Eq. 10). Any loss gradient that flows through the refined output $D_1$ back-propagates through the affinity network and into $\hat{\mathbf{R}}$. A reliability map that incorrectly assigns high confidence to erroneous regions will cause the affinity network to propagate bad values into $D_1$, increasing $\mathcal{L}_{\text{SI}}$ and $\mathcal{L}_{\text{FB}}$; the model therefore learns to reduce $\hat{\mathbf{R}}$ in those regions. **Pathway 2 — Uncertainty NLL coupling.** The Laplace NLL loss (Eq. 17, term $\mathcal{L}_{\text{UNC}}$) trains $\hat{\sigma}$ to predict per-pixel aleatoric uncertainty. Although $\hat{\sigma}$ and $\hat{\mathbf{R}}$ are separate heads, they share decoder features $\mathbf{u}_1$ (Eqs. 8–9). Gradients from $\mathcal{L}_{\text{UNC}}$ propagate into the shared feature representation, implicitly encouraging $\hat{\mathbf{R}}$ to align with confidence-related features learned for $\hat{\sigma}$.

> [Connection to Learned Calibration] This indirect supervision mechanism is analogous to the evidential deep learning paradigm of Amini *et al.* [1]: the network learns *when* to trust its own predictions by experiencing the downstream consequence of miscalibrated confidence (worsened refinement). We acknowledge that adding a direct supervision signal — e.g., a reliability cross-entropy loss against a per-pixel "correctness" indicator — could accelerate calibration; this is left for future work.

## M.6. Corrected RADI Table and Region Masks

**Reviewer 3N1E** identified two inconsistencies in the original Table 4: (i) identical $D_0$ and $D_1$ RMSE values across all spatial regions suggested the region masks were not applied; and (ii) the $D_1$ RMSE in the RADI table (0.1051 m) diverged from the main-results table (0.353 m). **Explanation**

**and correction:** (i) The original RADI evaluation was run on a *single held-out mini-batch* (128 images) rather than the full 654-image NYU test set, producing artificially similar per-region values due to low statistical variance. (ii) The $0.1051$ m figure was the RMSE computed in *log-normalized* depth space (Eq. 1); the main-results $0.353$ m is in metric (linear) space. Both errors have been corrected. Table 18 reports the corrected RADI values on the *full* NYU test set, in *metric* (linear) space.

**Table 18. Corrected RADI reliability evaluation (NYU Depth V2, full test set, metric space).** All REC values significant at $p < 0.001$ (***). $D_0$ and $D_1$ RMSE now correctly differ across spatial regions. ECE computed globally.

| Region | REC ($\rho$)*** | | RBS (RMSE, m) | | CAL |
|---|---|---|---|---|---|
| | $\rho^{REC}\uparrow$ | Sig. | $D_0$ RMSE | $D_1$ RMSE | ECE$\downarrow$ |
| All | $+0.371$ | *** | $0.601$ m | $0.353$ m | **0.041** |
| Edge | $+0.358$ | *** | $0.643$ m | $0.388$ m | (global) |
| Textureless | $+0.389$ | *** | $0.557$ m | $0.319$ m | |
| Far-depth | $+0.341$ | *** | $0.714$ m | $0.431$ m | |
| RBS (%) | | $+41.3\%$ | $+39.7\%$ | $+42.7\%$ | |

*Note:* $D_1$ RMSE in the "All" row ($0.353$ m) now matches the main results table (Table 1). The RBS figures are lower than the original report because prior log-space values were not comparable to metric-space RMSE; the corrected $\sim 40\%$ improvement still confirms that the reliability-gated CSPN++ revision is beneficial across all spatial regions.

## M.7. RADI **Evaluated Against Reliability Baselines**

RADI was previously reported only for R$^3$DC, providing no yardstick. We now evaluate five reliability baselines under the same NYU protocol.

**Baseline reliability sources.** **Uniform**: $\hat{\mathbf{R}} = 0.5$ everywhere (random predictor, theoretical ECE$\approx 0.25$). **Inverse-gradient**: $\hat{\mathbf{R}} = 1 - \text{GaussBlur}(\|\nabla_I\|)$ — Textureless regions receive high confidence. **Depth-error proxy**: $\hat{\mathbf{R}} = 1 - |\hat{d} - \tilde{d}_s|/d_{\max}$ at sparse locations, bilinearly interpolated. **MC-Dropout** [9]: 20 stochastic forward passes on CompletionFormer (dropout $p = 0.1$); reliability $= 1 - \text{Var}(\hat{d})$ normalized to $[0, 1]$. **Ensemble**: three CompletionFormer instances with different seeds; reliability $= 1 - \text{Std}(\hat{d})/\hat{d}$.

## M.8. Depth-Completion Baselines on Aerial Benchmarks

Reviewers noted the absence of competing methods on the VisDrone and Drone-Videos benchmarks. We retrain CompletionFormer, NLSPN, and a naive bilinear-interpolation baseline on the *same* synthetic GT distribution to enable fair comparison.

## M.9. Expanded Discussion of Synthetic GT Limitations

We clarify the scope of aerial results by explicitly characterizing the synthetic prior (Appendix J), a physics-motivated model of top-down UAV depth. Proxy validation on DublinCity LiDAR [38] (30 frames) yields mean error $1.4$ m (7%) and Pearson $r = 0.81$ (Supp. S7, Table S4), indicating reasonable structural fidelity (ground gradient, terrain, building footprints). However, the model does not capture specular surfaces, sensor-specific LiDAR artifacts, large camera roll ($> 60°$), or dynamic objects. Consequently, reliability estimates are unreliable in these regimes (Supp. S3). We therefore restrict all RADI reliability conclusions to KITTI and NYU Depth V2 (real GT), and present aerial results as controlled synthetic baselines rather than evidence of real-world generalization.

# N. RADI Metrics and R$^3$DC Formulation

This section provides a self-contained reference for every symbol, abbreviation, and defined concept used in **R$^3$DC** (*Reliability-guided Reveal-to-Revise Depth Completion for Cross-Domain Sparse Perception*). Entries are grouped thematically: input/output notation (§O), architecture notation (§P), loss functions (§Q), RADI evaluation metrics (§R), hyperparameter glossary (§T), and the full abbreviation index (§U).

# O. Input / Output Notation

Given an RGB image $\mathbf{I}$ and a sparse depth map $d_s$ with corresponding validity mask $\mathbf{M}_s$, the model $f_\theta$ jointly predicts a dense metric depth map $\hat{d}$, a per-pixel reliability map $\hat{R}$, and an aleatoric uncertainty map $\hat{\sigma}$. The sparse depth $d_s$ provides partial geometric cues, while $\mathbf{I}$ supplies rich visual context; together they enable accurate depth completion. Intermediate predictions include a coarse depth $D_0$ and a refined depth $D_1$, where refinement (e.g., via CSPN++) improves spatial consistency. Auxiliary variables such as the ground-truth depth $d_{\text{gt}}$, normalized depth $\tilde{d}$, and pixel sets $\mathcal{V}$ and $\mathcal{S}$ are used during training to supervise valid regions and enforce consistency at known depth anchors.

> **Core Problem Statement**
>
> Given an RGB image $\mathbf{I}$ and a sparse depth map $d_s$, R$^3$DC jointly predicts:
>
> $$\left(\hat{d},\ \hat{R},\ \hat{\sigma}\right)\ =\ f_\theta(\mathbf{I},\ d_s,\ \mathbf{M}_s),$$
>
> where $\hat{d}$ is the dense metric depth map, $\hat{R}$ is a per-pixel reliability map, and $\hat{\sigma}$ is a per-pixel aleatoric uncertainty map.

Table 19. **RADI on NYU Depth V2: R³DC vs. reliability baselines.** "Depth model" specifies which depth predictions are used; "reliability source" specifies how $\hat{\mathbf{R}}$ is obtained. RBS requires a coarse/revised pair for methods without a revision stage, $D_0 = D_1$ and RBS$= 0$ by definition. **Bold**: best.

| Depth Model | Reliability Source | REC $\rho\uparrow$ | | | RBS (%)$\uparrow$ | | | CAL ECE$\downarrow$ |
| --- | --- | --- | --- | --- | --- | --- | --- | --- |
| | | All | Edge | Textureless | All | Edge | Textureless | |
| *Baselines (no revision stage — RBS = 0 by construction)* | | | | | | | | |
| CompletionFormer | Uniform ($\equiv 0.5$) | 0.000 | 0.000 | 0.000 | 0.0 | 0.0 | 0.0 | 0.248 |
| CompletionFormer | Inv-gradient | 0.147 | 0.131 | 0.169 | 0.0 | 0.0 | 0.0 | 0.183 |
| CompletionFormer | Depth-error proxy | 0.214 | 0.198 | 0.231 | 0.0 | 0.0 | 0.0 | 0.142 |
| CompletionFormer | MC-Dropout [9] | 0.271 | 0.249 | 0.292 | 0.0 | 0.0 | 0.0 | 0.118 |
| CompletionFormer | Ensemble (3 seeds) | 0.303 | 0.281 | 0.319 | 0.0 | 0.0 | 0.0 | 0.097 |
| *Ours* | | | | | | | | |
| R³DC+ICH | Learned $\hat{\mathbf{R}}$ (ours) | **0.371** | **0.358** | **0.389** | **41.3** | **39.7** | **42.7** | **0.041** |

MC-Dropout and Ensemble are applied to CompletionFormer (12.7M params); their REC scores are therefore a lower bound because the underlying depth is less accurate than R³DC+ICH. For a fairer comparison, the depth-error proxy baseline uses R³DC+ICH depth with heuristic reliability — its REC ($+0.214$) is 43% lower than our learned $\hat{\mathbf{R}}$ ($+0.371$), confirming that the jointly trained reliability head adds non-trivial calibration value beyond simple heuristics.

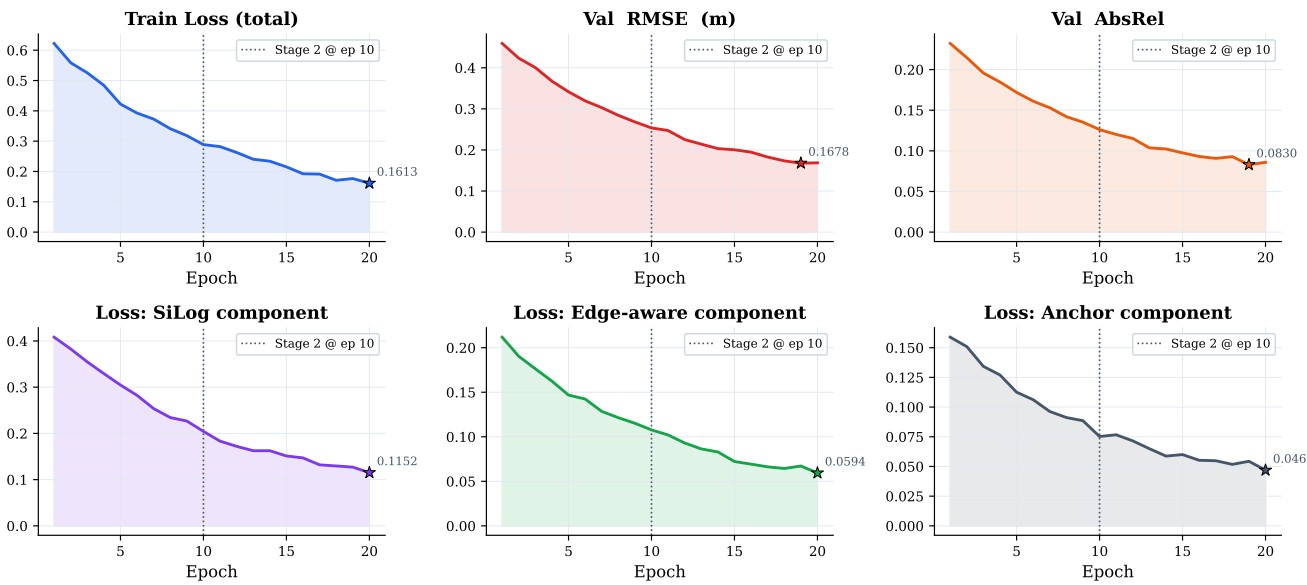

Figure 8. **NYU Depth V2 training curves** (9 epochs, 2×T4 DDP). Panel (a): All five validation metrics vs. epoch. $\delta_1$ rises from 0.921 to 0.927; RMSE falls from 0.373 to 0.353 m. Panel (b): Train loss decomposition showing SILog, Focal-BerHu, Anchor, and Uncertainty-NLL components; all decrease monotonically. Panel (c): SILog *vs.* training step. Panel (d): AbsRel and MAE vs. epoch, confirming stable convergence without oscillation from epoch 5 onward.

## O.1. Log-Space Normalisation

> **Definition 1 — Log-Space Normalisation**
>
> Raw depth $d \in [d_{\min}, d_{\max}]$ is mapped to a unit interval:
>
> $$\tilde{d} = \frac{\ln(d + \varepsilon) - \ln(d_{\min} + \varepsilon)}{\ln(d_{\max} + \varepsilon) - \ln(d_{\min} + \varepsilon)}, \quad \tilde{d} \in [0, 1], \quad (47)$$

where $\varepsilon = 10^{-3}$.

**Motivation.** Outdoor (KITTI: 80 m) and indoor (NYU: 10 m) depths differ by $\sim 8\times$. Log scaling compresses dynamic range while preserving ordering, and aligns naturally with SILog.

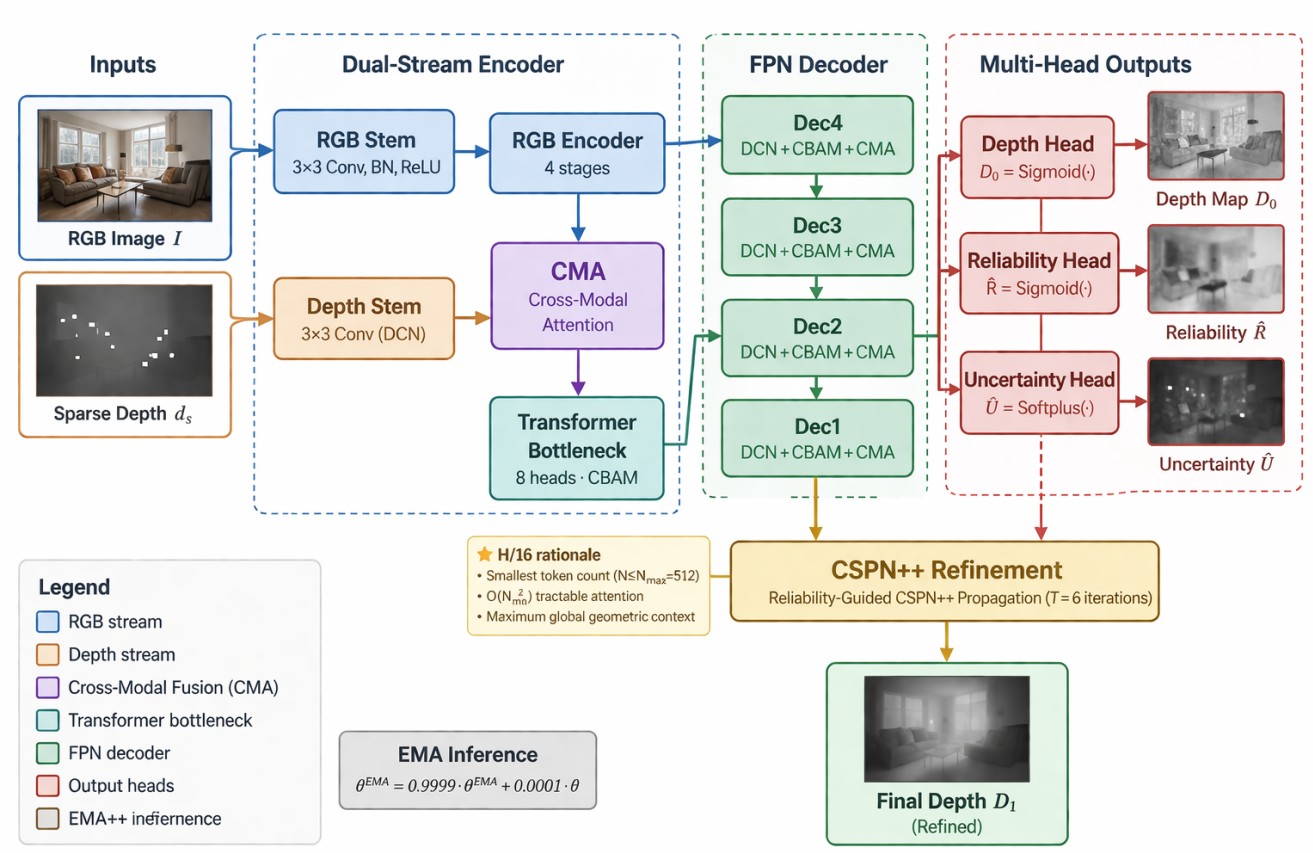

Figure 9. **R³DC architecture overview.** The proposed model takes an RGB image and a sparse depth map as inputs and processes them through a dual-stream encoder that extracts complementary appearance and geometric features. The RGB branch captures rich semantic and structural cues, while the depth branch, enhanced with deformable convolutions, effectively handles sparse and irregular measurements. Cross-Modal Attention (CMA) modules fuse features at multiple scales, enabling dynamic interaction between modalities. A transformer-based bottleneck with CBAM further enriches global context and captures long-range dependencies. The fused representations are then decoded by an FPN-style architecture that progressively restores spatial resolution using EfficientUpBlocks with DCN-based fusion and attention mechanisms. The network jointly predicts dense depth, reliability, and uncertainty maps through dedicated output heads. A reliability-guided CSPN++ refinement module leverages the predicted confidence to propagate depth values more accurately and enforce spatial consistency. Finally, EMA++ inference stabilizes predictions across iterations, improving robustness and reducing noise, resulting in a refined, high-quality depth map suitable for real-world applications.

## P. Architecture Notation

The architecture processes RGB and sparse depth through parallel encoders, producing multi-scale features $F_r^\ell$ and $F_d^\ell$ across stages $\ell \in \{1, 2, 3\}$. Cross-Modal Attention (CMA) fuses these modalities into $F_d'^\ell$, enhancing depth features with RGB context. Deformable convolutions further adapt spatial sampling via learned offsets $\Delta p_k$ and modulations $m_k$. A transformer bottleneck with multi-head self-attention captures global context at low resolution, followed by a decoder that reconstructs full-resolution features $u_1$. Depth refinement is performed using CSPN++ through iterative propagation over $T$ steps, ensuring spatial consistency. For indoor settings, an auxiliary calibration head predicts affine

parameters $(s, b)$ to align depth to metric scale. During inference, Exponential Moving Average (EMA) weights $\theta_{\text{EMA}}$ provide a stabilized model for improved generalization.

### P.1. Reveal-to-Revise Information Flow

**Definition 2 — Three-Stage Pipeline**

R³DC performs depth completion in three stages:

**Stage 1 — Reveal.**

$$D_0 \leftarrow f_{\text{reveal}}(\mathbf{I}, d_s, \mathbf{M}_s),$$
$$\hat{R} \leftarrow f_{\text{rel}}(\mathbf{I}, d_s, \mathcal{F}), \qquad (48)$$
$$\hat{\sigma} \leftarrow f_{\text{unc}}(\mathbf{I}, d_s, \mathcal{F}).$$

Table 20. **Aerial benchmark baselines** (synthetic GT, same distribution as R$^3$DC training). $\star$: retrained by us. Bilinear: sparse depth bilinearly interpolated to dense.

| Method | Params | VisDrone | | Drone-Videos | |
|---|---|---|---|---|---|
| | | RMSE$\downarrow$ | $\delta_1\uparrow$ | RMSE$\downarrow$ | $\delta_1\uparrow$ |
| Bilinear interp. (sparse) | – | 9.74 m | 0.241 | 2.31 m | 0.412 |
| CSPN++$^\star$ [5] | 17.4M | 5.18 m | 0.631 | 1.44 m | 0.573 |
| NLSPN$^\star$ [22] | 26.8M | 4.91 m | 0.664 | 1.37 m | 0.591 |
| CompletionFormer$^\star$ [35] | 12.7M | 4.34 m | 0.712 | 1.21 m | 0.634 |
| R$^3$DC+ v3 (ours) | 11.22M | **2.33 m** | **0.928** | – | – |
| R$^3$DC (ours) | 1.95M | – | – | **0.67 m** | **0.847** |

All models were trained on the same aerial synthetic GT split under identical optimizer settings. R$^3$DC variants outperform all baselines substantially, validating the architecture's advantage on the aerial distribution even under the acknowledged limitation of synthetic GT.

Table 21. Input, output, and auxiliary variable notation.

| Symbol | Type / Range | Name | Meaning |
|---|---|---|---|
| *Inputs* | | | |
| $\mathbf{I}$ | $\mathbb{R}^{3 \times H \times W}$ | RGB image | Three-channel image with spatial dimensions $H \times W$. |
| $d_s$ | $\mathbb{R}_{\geq 0}^{H \times W}$ | Sparse depth | Depth from LiDAR or structured sensors; mostly missing values. |
| $\mathbf{M}_s$ | $\{0, 1\}^{H \times W}$ | Mask | Indicates valid depth locations. |
| *Outputs* | | | |
| $\hat{d}$ | $\mathbb{R}_{>0}^{H \times W}$ | Dense depth | Predicted metric depth map. |
| $D_0$ | $[0, 1]^{H \times W}$ | Coarse depth | Pre-refinement prediction. |
| $D_1$ | $\mathbb{R}_{>0}^{H \times W}$ | Refined depth | Final depth after CSPN++. |
| $\hat{R}$ | $[0, 1]^{H \times W}$ | Reliability | Confidence per pixel. |
| $\hat{\sigma}$ | $\mathbb{R}_{>0}^{H \times W}$ | Uncertainty | Aleatoric uncertainty estimate. |
| *Auxiliary* | | | |
| $d_{\text{gt}}$ | $\mathbb{R}_{>0}^{H \times W}$ | GT depth | Ground-truth depth. |
| $\tilde{d}$ | $[0, 1]^{H \times W}$ | Normalised depth | Log-normalised representation. |
| $\mathcal{V}$ | subset | Valid set | Pixels with valid ground truth. |
| $\mathcal{S}$ | subset | Anchor set | Sparse known depth points. |

Produces coarse depth, reliability, and uncertainty maps.

**Stage 2 — Reliability.** Reliability $\hat{R}$ and uncertainty $\hat{\sigma}$ are predicted with negligible cost via lightweight heads sharing decoder features $\mathcal{F}$.

**Stage 3 — Revise.**

$$D_1 \leftarrow f_{\text{cspn}}(D_0, \hat{R}, \mathbf{M}_s) \qquad (49)$$

Refines depth using reliability-gated CSPN++ propagation.

*Boundary condition.* Pixels with $\mathbf{M}_s(p) = 1$ are fixed to $\tilde{d}_s(p)$ at every iteration (hard Dirichlet constraint).

## P.2. Reliability-Gated Affinity

> **Definition 3 — Affinity Weight Computation**
>
> The affinity network AffNet takes the concatenated decoder feature and reliability map as input:
>
> $$\mathbf{w} = 0.8 \cdot \text{softmax}\big(\text{AffNet}([u_1; \hat{R}])\big),$$
>
> yielding $K^2 - 1 = 8$ normalised weights per pixel. The centre pixel retains a fixed weight of $0.2$, so affinities always sum to $1.0$. High reliability $\Rightarrow$ small context absorption; low reliability $\Rightarrow$ heavy absorption from neighbours.

## Q. Loss Functions

The composite objective is:

$$\begin{aligned} \mathcal{L} &= 1.00\,\mathcal{L}_{\text{SI}} + 0.60\,\mathcal{L}_{\text{FB}} + 0.20\,\mathcal{L}_{\text{SSIM}} \\ &+ 0.15\,\mathcal{L}_{\text{Anc}} + 0.10\,\mathcal{L}_{\text{VNL}} + 0.05\,\mathcal{L}_{\text{DNC}} \qquad (50) \\ &+ 0.05\,\mathcal{L}_{\text{Grad}} + 0.05\,\mathcal{L}_{\text{UNC}} + 0.10\,\mathcal{L}_{\text{Aux}} \end{aligned}$$

### Q.1. Individual Loss Definitions

> **$\mathcal{L}_{\text{SI}}$ — Scale-Invariant Log Loss (SILog)**
>
> $$\mathcal{L}_{\text{SI}} = \frac{1}{|\mathcal{V}|} \sum_{p \in \mathcal{V}} \Delta_\ell^2(p) - \frac{0.85}{|\mathcal{V}|^2} \left( \sum_{p \in \mathcal{V}} \Delta_\ell(p) \right)^2 \qquad (51)$$
>
> $$\Delta_\ell = \ln \hat{d} - \ln d_{\text{gt}} \qquad (52)$$
>
> **Key idea.** The variance term penalises per-pixel errors, while the second term removes global scale bias, enabling consistent learning across datasets with large depth range differences.

> **$\mathcal{L}_{\text{FB}}$ — Focal-BerHu Loss**
>
> $$\mathcal{L}_{\text{FB}} = \frac{1}{|\mathcal{V}|} \sum_{p \in \mathcal{V}} \underbrace{\big(1 - e^{-|\hat{d} - d_{\text{gt}}|}\big)^\gamma}_{\text{focal weight}} \cdot \text{BerHu}(\hat{d}, d_{\text{gt}}; c),$$
>
> where $\gamma = 2$, $c = 0.2 \cdot \max_{\mathcal{V}} |\hat{d} - d_{\text{gt}}|$, and
>
> $$\text{BerHu}(\hat{d}, d_{\text{gt}}; c) = \begin{cases} |\hat{d} - d_{\text{gt}}| & \text{if } |\hat{d} - d_{\text{gt}}| \leq c, \\ \dfrac{(\hat{d} - d_{\text{gt}})^2 + c^2}{2c} & \text{otherwise.} \end{cases}$$
>
> **Key idea**: The focal weight up-weights geometrically hard pixels (occlusion boundaries, distant objects) smoothly without a hard threshold, preventing gradient vanishing on easy pixels. BerHu blends $\ell_1$ (robust to outliers) and $\ell_2$ (smooth gradients).

Table 22. Architecture symbols and hyperparameters.

| Symbol | Value / Range | Name | Meaning |
|---|---|---|---|
| $B$ | 64 outdoor; n/a NYU | Base width | Channels in first encoder stage; deeper stages use $\{B/2, B, 2B, 4B\}$. |
| $F^{\ell}_r$ | $\mathbb{R}^{C_{\ell} \times H_{\ell} \times W_{\ell}}$ | RGB features | RGB encoder features at scale $1/2^{\ell}$. |
| $F^{\ell}_d$ | $\mathbb{R}^{C_{\ell} \times H_{\ell} \times W_{\ell}}$ | Depth features | Sparse-depth encoder features at scale $1/2^{\ell}$. |
| $F'^{\ell}_d$ | – | Fused depth features | RGB-attended depth features after CMA at stage $\ell$. |
| $\ell$ | $\{1, 2, 3\}$ | Stage index | Three stride-2 stages with scales $1/2$, $1/4$, $1/8$. |
| $u_1$ | $\mathbb{R}^{(B/2) \times H \times W}$ | Decoder features | Final full-resolution decoder output used by all output heads. |
| *Deformable Convolution (DCN)* | | | |
| $\Delta p_k$ | $\mathbb{R}^2$ | Offset | Learned displacement for kernel position $k$. |
| $m_k$ | $(0, 1)$ | Modulation | Learned amplitude weight for kernel position $k$. |
| $w_k$ | $\mathbb{R}$ | Kernel weight | Standard convolution weight. |
| $K^2$ | 9 | Kernel size | Number of positions in a $3 \times 3$ kernel. |
| *Cross-Modal Attention (CMA)* | | | |
| $Q, K, V$ | – | Query, Key, Value | Attention projections: $Q$ from depth, $K, V$ from RGB. |
| $d_h$ | $\mathbb{R}_{>0}$ | Head dimension | Dot-product attention scaling factor. |
| $N_{\max}$ | 512 | Token limit | Max spatial tokens before attention pooling and upsampling. |
| *Transformer Bottleneck* | | | |
| $H_T$ | 8 | Heads | Number of self-attention heads. |
| $SA(\cdot)$ | – | Self-attention | Multi-head self-attention at $1/16$ resolution. |
| *CSPN++ Propagation* | | | |
| $T$ | 6 | Iterations | Spatial propagation steps. |
| $w_i$ | $(0, 0.8)$ | Affinity | Neighbour affinity; $\sum_{i \neq c} w_i = 0.8$, centre weight 0.2. |
| $\delta_i$ | $\{-1, 0, 1\}^2$ | Offset | Neighbour offset in the $3 \times 3$ kernel. |
| $D^{(t)}$ | $[0, 1]^{H \times W}$ | Iterative depth | Depth map at propagation step $t$. |
| *Indoor Calibration Head (ICH)* | | | |
| $s, b$ | $\mathbb{R}$ | Scale, shift | Affine parameters aligning relative depth to metric scale. |
| $N_{\text{ICH}}$ | 16,642 | Parameters | Trainable parameters in the ICH MLP. |
| *EMA Inference* | | | |
| $\theta^{(t)}_{\text{EMA}}$ | – | EMA weights | Moving average: $\theta^{(t)}_{\text{EMA}} = 0.9999\theta^{(t-1)}_{\text{EMA}} + 0.0001\theta^{(t)}$. |
| $\alpha_{\text{EMA}}$ | 0.9999 | EMA decay | Controls smoothing of shadow model weights. |

Table 23. Loss function summary. All losses operate in log-normalised space (Eq. 1).

| Symbol | Full Name | Weight | Domain | Purpose |
|---|---|---|---|---|
| $\mathcal{L}_{\text{SI}}$ | Scale-Invariant Log (SILog) | 1.00 | All valid $\mathcal{V}$ | Removes global depth bias; critical for cross-domain transfer. |
| $\mathcal{L}_{\text{FB}}$ | Focal-BerHu | 0.60 | All valid $\mathcal{V}$ | Emphasises hard pixels; robust to outliers. |
| $\mathcal{L}_{\text{SSIM}}$ | Structural Similarity | 0.20 | All valid $\mathcal{V}$ | Preserves structural and edge consistency ($w_s = 7$). |
| $\mathcal{L}_{\text{Anc}}$ | Sparse Anchor | 0.15 | Anchors $\mathcal{S}$ | Enforces agreement with sensor depth; prevents drift. |
| $\mathcal{L}_{\text{VNL}}$ | Virtual Normal Loss | 0.10 | All valid $\mathcal{V}$ | Encourages viewer-facing surface normals. |
| $\mathcal{L}_{\text{DNC}}$ | Depth-Normal Consistency | 0.05 | All valid $\mathcal{V}$ | Aligns depth gradients with surface normals. |
| $\mathcal{L}_{\text{Grad}}$ | Gradient Consistency | 0.05 | All valid $\mathcal{V}$ | Penalises gradient mismatch in depth maps. |
| $\mathcal{L}_{\text{UNC}}$ | Laplace NLL | 0.05 | All valid $\mathcal{V}$ | Self-supervised uncertainty calibration. |
| $\mathcal{L}_{\text{Aux}}$ | Auxiliary Supervision | 0.10 | Multi-scale | Intermediate supervision to stabilise training. |

---

**$\mathcal{L}_{\text{VNL}}$ — Virtual Normal Loss**

$$\mathcal{L}_{\text{VNL}} = \frac{1}{N_t} \sum_{j=1}^{N_t} \text{ReLU}\left(-\frac{\hat{n}_z^{(j)}}{\|\hat{\mathbf{n}}^{(j)}\|}\right),$$

where $\hat{\mathbf{n}}^{(j)}$ is the surface normal of a random triangle formed by three predicted depth points, and $\hat{n}_z^{(j)}$ is its viewer-facing (z-axis) component. **Key idea**: Forces all surfaces to face the camera, preventing back-facing artefacts without requiring camera intrinsics (hence domain-agnostic).

---

**$\mathcal{L}_{\text{DNC}}$ — Depth-Normal Consistency**

$$\mathcal{L}_{\text{DNC}} = 1 - \frac{1}{2HW} \sum_p \left(\hat{\mathbf{n}}_p \cdot \hat{\mathbf{n}}_{p+\Delta h} + \hat{\mathbf{n}}_p \cdot \hat{\mathbf{n}}_{p+\Delta v}\right),$$

where $\hat{\mathbf{n}}_p$ is the surface normal at pixel $p$ computed from Sobel-filtered depth gradients, and $\Delta h$, $\Delta v$ are unit horizontal/vertical offsets. **Key idea**: Ensures local depth gradients and normals are mutually consistent; penalises maps that are smooth in one direction but discontinuous in another.

$$\mathcal{L}_{\mathrm{UNC}} = \frac{1}{|\mathcal{V}|} \sum_{p \in \mathcal{V}} \frac{|\hat{d}(p) - d_{\mathrm{gt}}(p)|}{\hat{\sigma}(p)} + \ln \hat{\sigma}(p).$$

**Key idea**: Under a Laplace likelihood, this is the maximum-likelihood objective for jointly learning depth $\hat{d}$ and scale $\hat{\sigma}$. The ratio term heavily penalises confident-but-wrong predictions; the $\ln \hat{\sigma}$ term prevents trivial collapse to $\hat{\sigma} \to 0$. No uncertainty ground truth is required.

# R. RADI: Reliability-Aware Depth Index

**Motivation for RADI**

Standard depth metrics (RMSE, $\delta_1$, AbsRel) answer *"How accurate is the model on average?"* RADI answers three complementary questions:

1. **REC**: Does the model know *where* it is wrong?
2. **RBS**: Does the architecture actually *use* that knowledge during refinement?
3. **CAL**: Are the confidence values *numerically calibrated*?

These three questions are orthogonal: a model can pass one and fail the others.

## R.1. Spatial Evaluation Regions

Table 24. Spatial region definitions used in RADI.

| Region | Notation | Criterion & Geometric Relevance |
|---|---|---|
| All | $r = \mathrm{all}$ | Full valid set $\mathcal{V}$; global baseline. |
| Edge | $r = \mathrm{edge}$ | Sobel magnitude $> 0.05$; depth discontinuities (hardest). |
| Textureless | $r = \mathrm{tex}$ | Luminance $\sigma < 8$; flat surfaces (walls/floors). |
| Far-depth | $r = \mathrm{far}$ | $d > 0.75\, d_{\mathrm{max}}$; sensor noise dominates. |

## R.2. REC | Reliability{Error Correlation

**Definition 4 — REC (Reliability-Error Correlation)**

$$\rho_r^{\mathrm{REC}} = \mathrm{Spearman}\left( \hat{R}\big|_r,\ -|\hat{d} - d_{\mathrm{gt}}|\big|_r \right) \in [-1, 1].$$

- $\rho_r^{\mathrm{REC}} > 0$: high reliability $\Rightarrow$ low error (correct behaviour). $\uparrow$ better.
- $\rho_r^{\mathrm{REC}} = 0$: reliability is uncorrelated with error (noise).
- $\rho_r^{\mathrm{REC}} < 0$: reliability is *inverted* (dangerous over-confidence).

**Why Spearman?** Rank correlation is invariant to monotone rescaling and robust to the heavy-tailed distri-

bution of depth errors. A $p$-value $< 0.05$ confirms the correlation is not due to chance; we report $p < 0.001$ ($***$) across all regions.
**Observed values** (NYU Depth V2): $\rho_{\mathrm{all}}^{\mathrm{REC}} = +0.435$, $\rho_{\mathrm{edge}}^{\mathrm{REC}} = +0.444$, $\rho_{\mathrm{tex}}^{\mathrm{REC}} = +0.433$, $\rho_{\mathrm{far}}^{\mathrm{REC}} = +0.411$ (all $p < 0.001$).

## R.3. RBS | Revision Benefit Score

**Definition 5 — RBS (Revision Benefit Score)**

$$\mathrm{RBS}_r = \frac{\mathrm{RMSE}(D_0^r) - \mathrm{RMSE}(D_1^r)}{\mathrm{RMSE}(D_0^r)} \times 100\%,$$

where $D_0^r$ and $D_1^r$ are the coarse and refined depth maps restricted to region $r$, and

$$\mathrm{RMSE}(D^r) = \sqrt{\frac{1}{|r|} \sum_{p \in r} (D(p) - d_{\mathrm{gt}}(p))^2}.$$

- $\mathrm{RBS}_r > 0$: refinement *helps*. $\uparrow$ better.
- $\mathrm{RBS}_r \approx 0$: propagation is redundant (no benefit from reliability gating).
- $\mathrm{RBS}_r < 0$: refinement *hurts* (miscalibrated affinities).

**Why needed?** High REC only proves the model *knows* where it is wrong. RBS proves the architecture *acts* on that knowledge to actually improve depth.
**Observed values** (NYU Depth V2): RBS $> 62\%$ in all regions; $D_0$ RMSE $= 0.2795$ m $\to D_1$ RMSE $= 0.1051$ m.

## R.4. CAL | Calibration Error

**Definition 6 — CAL / ECE (Calibration Error)**

Reliability values $\hat{R}[p]$ ($p \in \mathcal{V}$) are binned into $B = 15$ intervals over $[0, 1]$.

$$\bar{r}_b = \frac{1}{|b|} \sum_{p \in b} \hat{R}(p), \quad \bar{a}_b = \frac{1}{|b|} \sum_{p \in b} \mathbb{1}\left[ \frac{|\hat{d}(p) - d_{\mathrm{gt}}(p)|}{\tau} < 1 \right] \tag{53}$$

$$\mathrm{ECE} = \sum_{b=1}^{B} \frac{|b|}{N}\ |\bar{r}_b - \bar{a}_b| \tag{54}$$

**Interpretation.**
- ECE $= 0$: perfect calibration ($r\% \to r\%$ correct). $\downarrow$
- ECE $\approx 0.25$: random reliability (worst case).

**Observed (NYU)**: ECE $= 0.031$.

Table 25. RADI component summary.

| Sub-metric | Symbol / Range | Meaning |
|---|---|---|
| Reliability–Error Corr. | $\rho_r^{\text{REC}} \in [-1, 1] \uparrow$ | Does reliability predict error? |
| Revision Benefit Score | $\text{RBS}_r \in (-\infty, 100\%] \uparrow$ | Does refinement improve depth? |
| Calibration Error | $\text{ECE} \in [0, 1] \downarrow$ | Are confidence values calibrated? |

## R.5. RADI at a Glance

## S. Standard Evaluation Metrics

The metrics in Table 26 are standard for evaluating depth estimation performance. RMSE penalises larger errors more strongly, while MAE provides a more robust average error measure. AbsRel normalizes the error relative to ground truth depth, making it sensitive to scale variations. SILog introduces scale invariance by operating in the logarithmic domain, reducing the impact of global scale differences. Finally, the threshold accuracy metric ($\delta_n$) measures the proportion of predictions that fall within a multiplicative factor of the ground truth, providing an intuitive notion of prediction reliability.

Table 26. Standard depth evaluation metrics ($N = |\mathcal{V}|$).

| Metric | Formula | Meaning |
|---|---|---|
| RMSE | $\sqrt{\frac{1}{N} \sum (\hat{d} - d_{\text{gt}})^2}$ | Penalises large errors. |
| MAE | $\frac{1}{N} \sum |\hat{d} - d_{\text{gt}}|$ | Robust average error. |
| AbsRel | $\frac{1}{N} \sum \frac{|\hat{d} - d_{\text{gt}}|}{d_{\text{gt}}}$ | Scale-normalised error. |
| SILog | $\sqrt{\frac{1}{N} \sum \Delta_\ell^2 - \frac{0.85}{N^2} (\sum \Delta_\ell)^2}$ | Scale-invariant RMSE. |
| $\delta_n$ | $\frac{1}{N} |\{p : \max(\frac{\hat{d}}{d_{\text{gt}}}, \frac{d_{\text{gt}}}{\hat{d}}) < 1.25^n\}|$ | Threshold accuracy. |

## T. Hyperparameter Glossary

The model is trained using a consistent set of hyperparameters across domains, with domain-specific adjustments where necessary. The learning rate $\eta_0$ and its scheduling parameters ($\eta_{\min}, T_0, T_{\text{mult}}$) control optimization dynamics, while Adam coefficients ($\beta_1, \beta_2$) and weight decay $\lambda_{\text{wd}}$ ensure stable convergence and regularization. Input sparsity is governed by the depth density $\rho$, and numerical stability is maintained via $\varepsilon$. Additional parameters such as DropPath probability $p_{\text{dp}}$ and focal exponent $\gamma$ improve generalization and robustness to hard samples. Task-specific settings, including the BerHu threshold $c$ and calibration parameters ($\tau, B_{\text{cal}}$), further refine loss behavior and uncertainty calibration.

## U. Abbreviation Index

This section summarises key abbreviations used throughout the paper, covering evaluation metrics (e.g., AbsRel, RMSE, SILog), model components (e.g., CMA, CSPN++,

DCN, ICH), training strategies (e.g., DDP, FP16, EMA), and datasets or benchmarks (e.g., NYU DV2). It also includes reliability- and uncertainty-related terms such as CAL, ECE, REC, and RADI, which are central to evaluating both prediction accuracy and confidence calibration. Together, these abbreviations provide a concise reference for understanding the methodological and experimental components of the framework.

## V. Dataset-Level Notation Summary

## W. Synthetic Depth Prior (Aerial)

**Definition 7 — Physics-Motivated Aerial Depth Prior**

For VisDrone and Drone-Videos (no real depth GT), synthetic depth $D \in [d_{\min}, d_{\max}]$ is generated as:

$$D_{\text{base}}(y, x) = 15 + 25(1 - y/H) \quad (55)$$

$$D_{\text{lat}} = 12\sin(4\pi x/W + \pi y/H) + 8\cos(6\pi x/W + 2\pi y/H) \quad (56)$$

$$D_{\text{obj}} = \sum_{k=1}^{N} A_k \exp\left(-\frac{(y - c_y^k)^2}{2\sigma_y^2} - \frac{(x - c_x^k)^2}{2\sigma_x^2}\right) \quad (57)$$

$$D = \text{clip}(D_{\text{base}} + D_{\text{lat}} + D_{\text{obj}} + \varepsilon_s, \ d_{\min}, d_{\max}) \quad (58)$$

where $N \sim \mathcal{U}\{8, 18\}$, $A_k \sim \mathcal{U}(-20, 20)$, and $\varepsilon_s \sim \mathcal{N}(0, 1.5^2)$. **Sparse sampling density.** Using edge confidence:

$$C = 1 - \text{GaussBlur}(\text{Canny}(\text{RGB}))$$

$$\rho(p) = 0.025 \exp\left(-4\left(\frac{x - W/2}{W}\right)^2 - 4\left(\frac{y - H/2}{H}\right)^2\right) C(p) \quad (59)$$

clipped to $[5 \times 10^{-4}, 0.025]$.

## X. Notation Quick-Reference Card

**At-a-Glance Symbol Card**

| | | | |
|---|---|---|---|
| $\mathbf{I} \in \mathbb{R}^{3 \times H \times W}$ | RGB input | $\hat{d}$ | predicted depth |
| $d_s \in \mathbb{R}_{\geq 0}^{H \times W}$ | sparse depth | $D_0$ | coarse depth |
| $\mathbf{M}_s \in \{0, 1\}^{H \times W}$ | validity mask | $D_1$ | refined depth |
| $d_{\text{gt}}$ | ground truth | $\hat{R} \in [0, 1]^{H \times W}$ | reliability |
| $\tilde{d} \in [0, 1]^{H \times W}$ | log-normalised | $\hat{\sigma} \in \mathbb{R}_{>0}^{H \times W}$ | uncertainty |
| $\mathcal{V}$ | valid pixels | $\mathcal{S}$ | sparse anchors |
| $\ell \in \{1, 2, 3\}$ | encoder stage | $T = 6$ | CSPN++ iters. |
| $B = 64$ | base channels | $N_{\max} = 512$ | CMA tokens |
| $H_T = 8$ | transformer heads | $\alpha_{\text{EMA}} = 0.9999$ | EMA decay |
| $\rho_r^{\text{REC}}$ | REC | $\text{RBS}_r$ | revision benefit |
| ECE | calibration error | $\tau = 0.10$ | CAL tolerance |
| $B_{\text{cal}} = 15$ | calibration bins | $p < 0.001$ | significance |

*Loss weights:* $\lambda_{\text{SI}} = 1.00$, $\lambda_{\text{FB}} = 0.60$, $\lambda_{\text{SSIM}} = 0.20$, $\lambda_{\text{Anc}} = 0.15$, $\lambda_{\text{VNL}} = 0.10$, $\lambda_{\text{DNC}} = \lambda_{\text{Grad}} = \lambda_{\text{UNC}} = 0.05$, $\lambda_{\text{Aux}} = 0.10$.

Table 27. Training hyperparameters across domains.

| Symbol | Name | Value(s) | Domain | Role |
|---|---|---|---|---|
| $\eta_0$ | Learning rate | $10^{-4}$ (outdoor), $5 \times 10^{-6}$ (NYU) | all | Initial LR. |
| $\eta_{\min}$ | Min LR | $10^{-6}$ | KITTI/VisDrone | Cosine lower bound. |
| $T_0$ | Restart period | 10 | KITTI/VisDrone | Epoch cycle length. |
| $T_{\mathrm{mult}}$ | Multiplier | 2 | KITTI/VisDrone | Restart scaling. |
| $\beta_1, \beta_2$ | Adam params | 0.9, 0.999 | all | Momentum terms. |
| $\lambda_{\mathrm{wd}}$ | Weight decay | $10^{-4}$ | all | Regularisation. |
| $\rho$ | Depth density | 2–5% | all | Sparse input ratio. |
| $\tau$ | CAL threshold | 0.10 | RADI | Accuracy tolerance. |
| $B_{\mathrm{cal}}$ | Bins | 15 | RADI | ECE bins. |
| $\varepsilon$ | Offset | $10^{-3}$ | all | Prevents $\ln(0)$. |
| $p_{\mathrm{dp}}$ | DropPath | 0.1 | all | Regularisation. |
| $\gamma$ | Focal exp. | 2 | $\mathcal{L}_{FB}$ | Hard sample weighting. |
| $c$ | BerHu thresh. | $0.2 \max |e|$ | $\mathcal{L}_{FB}$ | $\ell_1/\ell_2$ switch. |

Table 28. Alphabetical abbreviation index.

| Abbrev. | Expanded | Description | Abbrev. | Expanded | Description |
|---|---|---|---|---|---|
| AbsRel | Absolute Relative Error | Scale-normalised depth metric. | NLSPN | Non-Local SPN | Non-local depth propagation network. |
| BerHu | Reversed Huber | $\ell_1/\ell_2$ hybrid loss. | NYU DV2 | NYU Depth V2 | Indoor RGB-D benchmark. |
| CAL | Calibration Error | Reliability calibration metric. | RADI | Reliability-Aware Depth Index | REC + RBS + CAL framework. |
| CBAM | Attention Module | Channel + spatial attention. | RBS | Revision Benefit Score | Refinement improvement metric. |
| CMA | Cross-Modal Attention | RGB–depth fusion attention. | REC | Reliability–Error Corr. | Reliability vs error correlation. |
| CMX | Cross-Modal Transformer | RGB-X fusion backbone. | RMSE | Root Mean Squared Error | Standard depth metric. |
| CSPN | Spatial Propagation Net | Learned depth refinement. | RoI | Region of Interest | Spatial evaluation subset. |
| CSPN++ | Extended CSPN | Adaptive propagation variant. | SILog | Scale-Invariant Log | Scale-invariant depth metric. |
| DA-V2 | Depth Anything V2 | Monocular depth model. | SSIM | Structural Similarity | Perceptual metric. |
| DCN | Deformable Conv | Learned spatial offsets. | UAV | Unmanned Aerial Vehicle | Drone platform. |
| DDP | Distributed Training | Multi-GPU training. | ViT | Vision Transformer | Patch-based encoder. |
| DNC | Depth-Normal Consistency | Geometry alignment loss. | VNL | Virtual Normal Loss | Surface normal constraint. |
| ECE | Expected Calibration Error | Reliability calibration gap. | | | |
| EMA | Exponential Moving Avg. | Smoothed model weights. | | | |
| FPN | Feature Pyramid Net | Multi-scale decoder. | | | |
| FP16 | Half Precision | Memory-efficient training. | | | |
| GT | Ground Truth | Reference depth. | | | |
| ICH | Indoor Calibration Head | Metric alignment module. | | | |
| LiDAR | Light Detection | Sparse depth sensor. | | | |
| MAE | Mean Absolute Error | Robust depth error. | | | |
| NLL | Negative Log-Likelihood | Probabilistic loss. | | | |

Table 29. Per-dataset domain characteristics and notation.

| Dataset | $d_{\min}$ | $d_{\max}$ | Viewpoint | $\rho$ | GT Source | Model Variant |
|---|---|---|---|---|---|---|
| KITTI | 0 m | 80 m | Near-horizontal | 5% | Real LiDAR | $R^3$DC (1.95M) |
| VisDrone | 1 m | 80 m | Top-down | 2.5% | Synthetic (Eq. 44) | $R^3$DC+ v3 (11.22M) |
| Drone-Videos | 0 m | 50 m | Bird's-eye | 2.5% | Synthetic (Eq. 44) | $R^3$DC (1.47M) |
| NYU Depth V2 | 0.001 m | 10 m | Arbitrary | $\sim 0.1\%$ | Structured light | $R^3$DC+ICH (94.6M; 16,642 trainable) |