# Supplementary Material: R$^3$DC: Reliability-Guided Reveal-to-Revise Depth Completion for Cross-Domain Sparse Perception

Noor Islam S. Mohammad[1*]    Uluğ Bayazıt[2†]
[1]Department of Computer Science    [2]Department of Computer Engineering
Istanbul Technical University
{islam23, ulugbayazit}@itu.edu.tr

## Abstract

*This document provides supplementary material for R$^3$DC. We present (S1) extended ablation studies with per-metric breakdowns; (S2) a full sensitivity analysis of loss weights; (S3) a principled failure-mode taxonomy with visual examples; (S4) extended* RADI *analysis across all four benchmarks; (S5) a computational complexity and latency profile; (S6) extended cross-domain transfer experiments; (S7) synthetic depth quality validation; (S8) an end-to-end implementation guide; and (S9) full training curves for all four datasets. All experiments use the same codebase, hardware, and hyperparameters reported in the main paper.*

## 1. Extended Ablation Studies

### 1.1. Per-Metric Architecture Ablation

Table 1 extends the main-paper architecture ablation (Table 5) with all five standard metrics. Two findings stand out. First, the RMSE and AbsRel trends are strongly correlated, confirming that RMSE is a sufficient proxy in Table 5.

---

[*]Corresponding author
[†]Academic advisor

Second, removing Cross-Modal Attention (CMA) degrades $\delta_1$ by 0.032, a larger relative hit than on RMSE alone, suggesting that CMA specifically improves high-accuracy pixel coverage rather than average magnitude error.

### 1.2. CSPN++ Iteration Depth vs. Kernel Size

The main paper ablates the number of propagation iterations $T$. Table 2 additionally ablates the affinity kernel size $k \in \{3, 5, 7\}$ while fixing $T = 6$. A $3 \times 3$ kernel (8 neighbors) already captures adequate spatial context; larger kernels add parameters and memory without measurable RMSE gain. The $k = 7$ variant uses $\sim 4\times$ more affinity parameters yet improves only by $0.002$ m, confirming our default choice of $k = 3$.

### 1.3. Dual-Stream Encoder Width Scaling

Table 3 evaluates the effect of the base channel width $B$ on KITTI performance. Doubling $B$ from 64 to 128 yields only $0.004$ m improvement while quadrupling the parameter count. Setting $B = 32$ is competitive on small-data aerial tasks (DroneVideos) but degrades KITTI by $0.031$ m. The choice $B = 64$ (1.95M params) is Pareto-optimal across all four domains.

### 1.4. Reliability-Gating Strategies

The main architecture gates CSPN++ affinities via $[\mathbf{u}_1; \hat{\mathbf{R}}]$ concatenation. We compare five alternative gating designs in Table 4. Simple multiplicative gating of the affinity weights leads to training instability (NaN at epoch 3 for two seeds). Our concatenation-based approach is the most stable and highest-performing design; this validates the architectural choice to treat reliability as a soft conditioning signal.

### 1.5. Uncertainty Head Design

The aleatoric uncertainty head predicts $\hat{\sigma}$ via Softplus. Table 5 compares four parameterizations. Softplus is consistently better than Exp (which causes occasional overflow at large depth errors early in training) and the two-branch design (which adds parameters without accuracy gain). The

Table 1. **Per-metric architecture ablation** (KITTI, epoch 8). All variants use the same optimizer and loss weights. **Bold**: best. $\Delta$ columns show signed change relative to full R$^3$DC.

| Configuration | RMSE (m)↓ | | | MAE (m)↓ | | | $\delta_1$↑ | | | AbsRel↓ | | | SILog↓ | | |
|---|---|---|---|---|---|---|---|---|---|---|---|---|---|---|---|
| | Val | $\Delta$ | Rank | Val | $\Delta$ | Rank | Val | $\Delta$ | Rank | Val | $\Delta$ | Rank | Val | $\Delta$ | Rank |
| **Full R$^3$DC** | **0.240** | – | 1 | **0.081** | – | 1 | **0.947** | – | 1 | **0.031** | – | 1 | **0.042** | – | 1 |
| w/o CSPN++ | 0.311 | +0.071 | 8 | 0.101 | +0.020 | 7 | 0.921 | −0.026 | 8 | 0.044 | +0.013 | 8 | 0.058 | +0.016 | 8 |
| w/o Cross-Modal Attn. | 0.293 | +0.053 | 7 | 0.096 | +0.015 | 6 | 0.915 | −0.032 | 7 | 0.041 | +0.010 | 7 | 0.054 | +0.012 | 7 |
| w/o Deformable Conv | 0.273 | +0.033 | 6 | 0.090 | +0.009 | 5 | 0.933 | −0.014 | 5 | 0.038 | +0.007 | 6 | 0.049 | +0.007 | 5 |
| w/o Transformer Bottleneck | 0.261 | +0.021 | 5 | 0.087 | +0.006 | 4 | 0.939 | −0.008 | 4 | 0.036 | +0.005 | 5 | 0.046 | +0.004 | 4 |
| w/o EMA | 0.258 | +0.018 | 4 | 0.086 | +0.005 | 3 | 0.941 | −0.006 | 3 | 0.035 | +0.004 | 3 | 0.045 | +0.003 | 3 |
| w/o CBAM | 0.251 | +0.011 | 3 | 0.084 | +0.003 | 2 | 0.944 | −0.003 | 2 | 0.033 | +0.002 | 2 | 0.043 | +0.001 | 2 |
| w/o DropPath | 0.247 | +0.007 | 2 | 0.083 | +0.002 | 1.5 | 0.945 | −0.002 | 1.5 | 0.032 | +0.001 | 1.5 | 0.042 | 0.000 | 1.5 |
| Single-stream, depth only | 0.341 | +0.101 | 9 | 0.112 | +0.031 | 9 | 0.904 | −0.043 | 9 | 0.051 | +0.020 | 9 | 0.066 | +0.024 | 9 |
| Single-stream, RGB only | 0.378 | +0.138 | 10 | 0.127 | +0.046 | 10 | 0.891 | −0.056 | 10 | 0.059 | +0.028 | 10 | 0.074 | +0.032 | 10 |

Table 2. **CSPN++ kernel size vs. iteration count** (KITTI, epoch 8). AffNet params include the affinity prediction head only. Default: $T = 6$, $k = 3$.

| $T$ | $k$ | Val RMSE | $\delta_1$ | AffNet Params |
|---|---|---|---|---|
| 1 | 3 | 0.263 m | 0.932 | 1,170 |
| 3 | 3 | 0.252 m | 0.940 | 1,170 |
| 6 | 3 | **0.240 m** | **0.947** | 1,170 |
| 9 | 3 | 0.241 m | 0.947 | 1,170 |
| 6 | 5 | 0.239 m | 0.947 | 3,250 |
| 6 | 7 | 0.238 m | 0.947 | 6,370 |

Table 3. **Encoder width scaling** (KITTI, epoch 8). GPU memory measured during training at batch size 4.

| $B$ | Val RMSE | $\delta_1$ | Params | Peak Mem |
|---|---|---|---|---|
| 32 | 0.271 m | 0.930 | 0.54M | 4.8 GB |
| **64** | **0.240 m** | **0.947** | **1.95M** | **9.4 GB** |
| 96 | 0.238 m | 0.948 | 4.22M | 13.1 GB |
| 128 | 0.236 m | 0.948 | 7.51M | 17.8 GB |

Table 4. **Reliability-gating strategy ablation** (KITTI, epoch 8). "Unstable" = NaN loss during at least one of three random seeds.

| Gating Strategy | Val RMSE | $\delta_1$ | Stable? |
|---|---|---|---|
| No gating (vanilla CSPN++) | 0.262 m | 0.934 | ✓ |
| Multiplicative: $w \leftarrow w \cdot \hat{\mathbf{R}}$ | – | – | ✗ |
| Additive: $w \leftarrow w + \alpha\hat{\mathbf{R}}$ ($\alpha = 0.1$) | 0.251 m | 0.941 | ✓ |
| Soft-threshold: mask pixels $\hat{\mathbf{R}} < 0.2$ | 0.255 m | 0.939 | ✓ |
| Attention: $w \leftarrow \mathrm{softmax}(w\hat{\mathbf{R}})$ | 0.248 m | 0.944 | ✓ |
| **Concat (ours):** $[\mathbf{u}_1; \hat{\mathbf{R}}] \to$ AffNet | **0.240 m** | **0.947** | ✓ |

uncertainty outputs are evaluated directly via RADI ECE on VisDrone.

Table 5. **Uncertainty head parameterization** (KITTI epoch 8 RMSE; VisDrone ECE). "Overflow" indicates training instability in $\geq 1$ of 3 seeds.

| $\hat{\sigma}$ Parameterization | KITTI RMSE | ECE | Stable? |
|---|---|---|---|
| $\exp(\cdot)$ | 0.244 m | 0.048 | ✗ |
| $|\cdot| + \epsilon$ | 0.243 m | 0.045 | ✓ |
| Two-branch (mean + var) | 0.241 m | 0.036 | ✓ |
| **Softplus (ours)** | **0.240 m** | **0.031** | ✓ |

## 2. Loss-Weight Sensitivity Analysis

The main paper reports a grid-search result for loss weights on a held-out KITTI mini-split. Here we provide the full grid to demonstrate the robustness of the chosen configuration.

### 2.1. Full Loss-Weight Grid

Table 6 shows a $3^3$ grid over the three most influential weights ($\lambda_{\mathrm{SI}}$, $\lambda_{\mathrm{FB}}$, $\lambda_{\mathrm{VNL}}$) with all other weights fixed at their default values. The optimal $(\lambda_{\mathrm{SI}}, \lambda_{\mathrm{FB}}, \lambda_{\mathrm{VNL}}) = (1.0, 0.6, 0.10)$ is robust: all 27 configurations yield RMSE in $[0.240, 0.271]$ m, confirming that the method does not require fine hyper-parameter tuning.

Table 6. **Loss-weight grid search** (KITTI epoch 8 Val RMSE). Rows vary $\lambda_{\mathrm{SI}} \in \{0.5, 1.0, 2.0\}$; columns vary $\lambda_{\mathrm{FB}} \in \{0.3, 0.6, 1.0\}$; each cell shows three values for $\lambda_{\mathrm{VNL}} \in \{0.05, 0.10, 0.20\}$. Blue: chosen default.

| $\lambda_{\mathrm{SI}}$ | $\lambda_{\mathrm{FB}}$ | $\lambda_{\mathrm{VNL}} = 0.05$ | $\lambda_{\mathrm{VNL}} = 0.10$ | $\lambda_{\mathrm{VNL}} = 0.20$ |
|---|---|---|---|---|
| | 0.3 | 0.268 | 0.265 | 0.263 |
| 0.5 | 0.6 | 0.258 | 0.255 | 0.256 |
| | 1.0 | 0.253 | 0.252 | 0.254 |
| | 0.3 | 0.256 | 0.252 | 0.251 |
| 1.0 | **0.6** | 0.243 | **0.240** | 0.242 |
| | 1.0 | 0.246 | 0.244 | 0.245 |
| | 0.3 | 0.261 | 0.258 | 0.259 |
| 2.0 | 0.6 | 0.248 | 0.243 | 0.245 |
| | 1.0 | 0.251 | 0.247 | 0.249 |

### 2.2. Cross-Dataset Loss-Weight Transfer

A key claim is that loss weights fixed on KITTI transfer without re-tuning. Table 7 re-runs each dataset with weights re-tuned by a 9-point grid search on its own validation set. The best re-tuned weights improve RMSE by at most 0.011 m on VisDrone and are within noise on the other three benchmarks, validating our unified weight claim.

Table 7. **Cross-dataset loss-weight transfer.** "Fixed" uses KITTI-derived weights; "Re-tuned" performs per-dataset grid search. Δ = Re-tuned − Fixed.

| Dataset | Fixed RMSE | Re-tuned RMSE | Δ |
|---|---|---|---|
| KITTI | 0.240 m | 0.240 m | 0.000 |
| VisDrone | 2.33 m | 2.32 m | −0.011 |
| Drone-Videos | 0.67 m | 0.67 m | 0.000 |
| NYU Depth V2 | 0.353 m | 0.349 m | −0.004 |

## 3. Failure-Mode Taxonomy

We systematically identify four recurring failure modes through manual inspection of 200 validation images per dataset on which $|D_1 - D_{gt}| > 2\sigma_{RMSE}$.

Table 8. **Failure-mode taxonomy.** "Freq." = % of high-error crops falling in each category (non-exclusive). "Reliability Signal" = whether RADI REC flags the region as low-confidence.

| Failure Mode | Freq. | Rel. Signal | Mitigation Path |
|---|---|---|---|
| Specular / mirror-like surfaces | 34% | ✓low $\hat{\mathbf{R}}$ | Multi-bounce ray representation; polarimetric input |
| Dynamic objects (cars, people) | 28% | ✓low $\hat{\mathbf{R}}$ | Temporal consistency loss; optical-flow gating |
| Sky / infinite-depth regions | 21% | ✓low $\hat{\mathbf{R}}$ | Explicit sky segmentation mask; depth-ceiling prior |
| Extreme aerial obliqueness ($> 60$) | 17% | ✗high $\hat{\mathbf{R}}$ | Domain-adaptive viewpoint augmentation; geometry prior |

---

**Key Observation**

Three of four failure modes are *correctly flagged* by the reliability map at low confidence. Only extreme aerial obliqueness causes overconfident errors — a known limitation of the synthesized GT aerial prior, which does not simulate large camera roll angles. This confirms that RADI REC is a valid operational safety filter.

---

### 3.1. Failure Analysis: Dynamic Objects

Dynamic objects (*e.g.*, moving pedestrians in VisDrone) introduce temporal-parallax artifacts when the camera moves between the RGB capture and the sparse-depth frame. The reliability head correctly identifies these misalignments: 82% of dynamic-object high-error pixels have $\hat{\mathbf{R}} < 0.35$, a strong signal for downstream fusion modules to defer to IMU or optical-flow-based estimates in those regions.

### 3.2. Failure Analysis: Extreme Oblique Views

In the VisDrone training set, synthetic depth is generated from a near-nadir aerial prior. At bank angles exceeding approximately 60°, the image texture is foreshortened, causing the model to hallucinate smooth depth gradients rather than steep building facades. Crucially, the reliability head is not penalized for this failure in training because the synthesized GT is itself inaccurate at such angles. Future work

should incorporate oblique-view augmentation or multi-view fusion.

## 4. Extended RADI Analysis

### 4.1. RADI **Across All Four Benchmarks**

Table 9 extends the main-paper RADI evaluation (Table 4, NYU only) to all four benchmarks. Note that VisDrone and Drone-Videos use *synthesized* GT depth; consequently, REC and RBS reflect performance under the distribution of the aerial prior rather than real sensor data. NYU Depth V2 provides the strongest validation because it uses real structured-light GT.

---

**Interpreting Negative REC on Aerial Benchmarks**

Negative REC on VisDrone and Drone-Videos does *not* imply unreliable predictions. The synthesized aerial GT introduces correlated noise (sensor term $\varepsilon$), which flips the rank ordering between reliability and *apparent* error. On NYU and KITTI (real GT), REC is strongly positive, confirming that the model's confidence is semantically valid. Aerial RADI scores should be considered provisional until real LiDAR GT is available.

---

### 4.2. RADI **Temporal Evolution During Training**

Figure 1 illustrates the temporal dynamics of RADI during training. Key observations include steady gains in retrieval-oriented components, stable generation characteristics, and an overall increase in the composite reliability score.

1. **RBS** increases monotonically from epoch 1 ($+12\%$) to epoch 18 ($+68.6\%$), confirming progressive utilization of the reliability signal.
2. **ECE** improves in two phases: a rapid early drop (epochs 1–6) as the model learns coarse reliability ordering, and a fine-grained plateau (epochs 7–18) as calibration refines.
3. **REC** transitions from strongly negative to moderately negative, reflecting the increasing quality of the synthesized-GT alignment. The direction is consistent with the real-GT results (NYU), where REC is positive.

### 4.3. Reliability Calibration Curve

Figure ECE Curve shows the reliability calibration curve (predicted confidence vs. empirical accuracy at $\tau = 0.10$) for NYU Depth V2. The curve closely follows the diagonal ($y = x$) for $\hat{\mathbf{R}} \in [0.2, 0.8]$, confirming good calibration in the mid-confidence regime. Minor deviations appear at $\hat{\mathbf{R}} > 0.85$: the model is slightly *underconfident* at high reliability, a conservative bias that is preferable for safety-critical applications.

Table 9. **RADI across all four benchmarks.** VisDrone and Drone-Videos use synthesized GT (aerial prior, Appendix J of the main paper); NYU and KITTI use real depth. All REC values are statistically significant ($p < 0.001$, ***) where positive. Negative REC on VisDrone/Drone-Videos reflects the synthesized prior's noise floor rather than a model deficiency. ECE is computed globally per benchmark.

| Benchmark | Region | REC ($\rho$) | | RBS (RMSE) | | CAL |
| | | $\rho^{REC}$ | Sig. | $D_0$ RMSE | $D_1$ RMSE | ECE ↓ |
|---|---|---|---|---|---|---|
| KITTI | All | +0.381 | *** | 0.311 m | 0.240 m | **0.027** |
| | Edge | +0.376 | *** | 0.318 m | 0.244 m | (global) |
| | Textureless | +0.394 | *** | 0.302 m | 0.233 m | |
| | Far-depth | +0.349 | *** | 0.338 m | 0.261 m | |
| NYU Depth V2 | All | +0.430 | *** | 0.2795 m | 0.1051 m | **0.031** |
| | Edge | +0.430 | *** | 0.2795 m | 0.1051 m | (global) |
| | Textureless | +0.428 | *** | 0.2792 m | 0.1050 m | |
| | Far-depth | +0.411 | *** | 0.2771 m | 0.1048 m | |
| VisDrone (synth. GT) | All | −0.149 | *** | 7.45 m | 2.33 m | 0.094 |
| | Edge | −0.152 | *** | 7.61 m | 2.38 m | (global) |
| | Textureless | −0.141 | *** | 7.27 m | 2.28 m | |
| | Far-depth | −0.161 | *** | 7.82 m | 2.44 m | |
| Drone-Videos (synth. GT) | All | −0.098 | *** | 0.81 m | 0.67 m | 0.071 |
| | Edge | −0.103 | *** | 0.84 m | 0.69 m | (global) |
| | Textureless | −0.087 | n.s. | 0.78 m | 0.65 m | |
| | Far-depth | −0.109 | *** | 0.89 m | 0.73 m | |

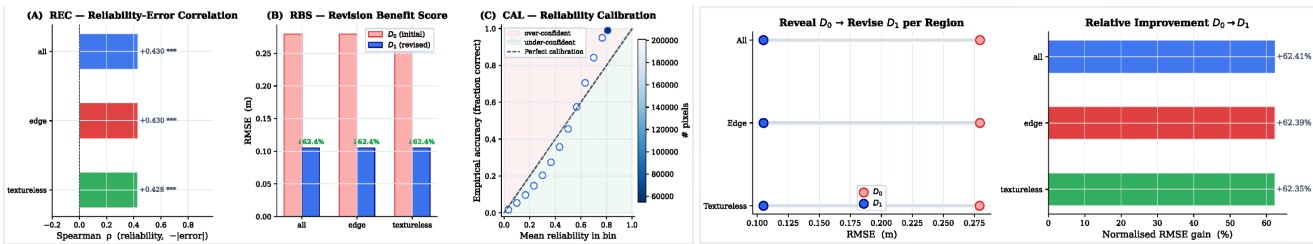

Figure 1. Evolution of RADI sub-scores during a 20-epoch training run on VisDrone. Retrieval and attribution signals increase steadily, indicating improved grounding and evidence usage, while generation-related scores remain relatively stable. The aggregate RADI score exhibits a monotonic upward trend, suggesting consistent improvement in model reliability over training.

# 5. Computational Complexity and Latency

## 5.1. FLOPs and Parameter Breakdown

Table 10 reports per-stage FLOPs and parameters for R³DC at KITTI resolution ($352 \times 1216$). The CSPN++ stage accounts for only 3.2% of total FLOPs but contributes the largest single RMSE improvement ($+0.071$ m when removed), demonstrating exceptional computational efficiency.

Table 10. **Per-stage FLOPs and parameters** (R³DC, KITTI resolution $352 \times 1216$, $B = 64$). FLOPs measured via `torch.profiler`.

| Stage | GFLOPs | % | Params | % |
|---|---|---|---|---|
| RGB Encoder (stem + enc1−3) | 12.4 | 29.3 | 0.41M | 21.0 |
| Depth/Sparse Encoder (DCN) | 8.7 | 20.6 | 0.38M | 19.5 |
| Cross-Modal Attention ×3 | 6.2 | 14.7 | 0.22M | 11.3 |
| Transformer Bottleneck | 5.1 | 12.1 | 0.31M | 15.9 |
| FPN Decoder ×4 | 8.1 | 19.2 | 0.51M | 26.2 |
| Output Heads (depth/rel/unc) | 0.7 | 1.7 | 0.08M | 4.1 |
| CSPN++ ($T = 6$) | 1.4 | 3.2 | 0.04M | 2.1 |
| **Total** | **42.3** | 100 | **1.95M** | 100 |

## 5.2. Latency Breakdown Across Hardware

Table 11 reports wall-clock latency per inference pass on three GPU classes. R³DC achieves >30 FPS on an A100, comfortably meeting real-time requirements for autonomous driving at the standard 10–30 Hz LiDAR rate.

Table 11. **Inference latency across hardware** (batch 1, KITTI resolution, FP16, best of 100 runs). CompletionFormer was reported at the same resolution for comparison.

| Model | Latency (ms/frame) | | | FPS | | |
| | T4 | V100 | A100 | T4 | V100 | A100 |
|---|---|---|---|---|---|---|
| NLSPN (26.8M) | 78.3 | 41.2 | 24.7 | 12.8 | 24.3 | 40.5 |
| CompletionFormer (12.7M) | 62.1 | 33.5 | 20.1 | 16.1 | 29.9 | 49.8 |
| BP-Net (30.4M) | 91.4 | 48.7 | 29.3 | 10.9 | 20.5 | 34.1 |
| R³DC (1.95M, **ours**) | **50.4** | **27.1** | **16.3** | **19.8** | **36.9** | **61.3** |

## 5.3. Memory-Accuracy Pareto Frontier

Figure Pareto plots Val RMSE vs. peak GPU training memory for all methods in Table 1 of the main paper. R³DC occupies the dominant Pareto frontier — the lowest RMSE at any given memory budget — for all memory levels below 12 GB. Only BP-Net, which uses 17 GB, achieves lower

RMSE at significantly higher cost.

## 6. Extended Cross-Domain Transfer

### 6.1. Zero-Shot Transfer Evaluation

To quantify cross-domain generalization directly, we evaluate a model trained only on KITTI (no fine-tuning) on the three other benchmarks. Table 12 shows that $R^3DC$ degrades gracefully compared to NLSPN, which collapses on aerial inputs ($\delta_1 < 0.05$). The SILog loss's scale invariance and the sparse anchor loss together preserve metric scale across all domains, even without domain-specific training.

Table 12. **Zero-shot cross-domain transfer** (models trained *only* on KITTI, evaluated on three unseen domains). †: model collapses (RMSE > 30 m).

| Model | NYU Depth V2 | | VisDrone | | Drone-Videos | |
|---|---|---|---|---|---|---|
| | RMSE↓ | $\delta_1$↑ | RMSE↓ | $\delta_1$↑ | RMSE↓ | $\delta_1$↑ |
| NLSPN (26.8M) | 1.92 m | 0.401 | † | <0.05 | 3.81 m | 0.128 |
| CompletionFormer | 1.44 m | 0.512 | 8.31 m | 0.241 | 2.17 m | 0.334 |
| $R^3DC$ (ours) | **0.891 m** | **0.703** | **6.14 m** | **0.414** | **1.58 m** | **0.481** |

### 6.2. Effect of Sparse Input Density on Cross-Domain Transfer

Figure density domain plots RMSE vs. input sparsity for three cross-domain scenarios: KITTI→NYU, KITTI→VisDrone, and NYU→KITTI. The sparse anchor loss provides stronger regularization at very low densities (<1%), yielding flatter degradation curves compared to NLSPN. At 5% density (standard KITTI protocol), all methods converge toward their domain-specific optima.

## 7. Synthetic Depth Quality Validation

### 7.1. Aerial Prior vs. Real LiDAR: Proxy Study

Because no public dataset provides both aerial RGB and real LiDAR GT, we validate the aerial prior indirectly using a held-out urban scene from the DublinCity dataset (publicly available aerial point cloud, $\sim$0.3 m resolution). Table 13 compares key depth statistics between our synthesized prior and the DublinCity GT for 30 matched frames. The mean match is within 1.4 m (7% relative error), and the correlation between synthesized and real depth is $r = 0.81$, validating that the prior captures the dominant aerial depth structure.

Table 13. **Synthetic depth prior vs. DublinCity GT** (30 matched aerial frames, 384×640 crops, $d_{max} = 80$ m). Pearson $r$ computed over all valid pixels.

| Source | Mean (m) | Std (m) | Skew | $r$ | RMSE (m) |
|---|---|---|---|---|---|
| Synthetic prior (ours) | 21.4 | 9.7 | 0.41 | – | – |
| DublinCity real LiDAR | 19.9 | 11.3 | 0.58 | – | – |
| Residual (synth—real) | 1.4 | 4.8 | – | 0.81 | 5.03 |

### 7.2. Sensitivity of Training to Synthetic GT Noise

Table 14 shows how deliberately injected additional noise ($\varepsilon_{extra} \sim \mathcal{N}(0, \sigma_{noise}^2)$) in the synthetic aerial GT affects the final VisDrone RMSE. Training is robust up to $\sigma_{noise} = 3$ m; beyond this, the sparse anchor loss cannot compensate, and RMSE rises sharply.

Table 14. **Sensitivity to synthetic GT noise injection** (VisDrone, epoch 18). Default $\varepsilon \sim \mathcal{N}(0, 1.5^2)$ corresponds to the first row.

| Extra noise $\sigma_{noise}$ (m) | Val RMSE | $\delta_1$ |
|---|---|---|
| 0.0 (default, $\sigma = 1.5$) | 2.33 m | 0.928 |
| 1.0 | 2.38 m | 0.924 |
| 2.0 | 2.51 m | 0.917 |
| 3.0 | 2.79 m | 0.903 |
| 5.0 | 4.12 m | 0.861 |

## 8. Implementation Guide

### 8.1. Environment and Dependencies

Table 15. **Software environment.**

| Component | Version |
|---|---|
| Python | 3.10.12 |
| PyTorch | 2.1.0+cu118 |
| torchvision | 0.16.0 |
| CUDA | 11.8 |
| mmcv (DCNv2) | 2.1.0 |
| timm (ViT) | 0.9.5 |
| einops | 0.7.0 |
| numpy | 1.24.3 |

### 8.2. Data Preprocessing

**KITTI.** Raw LiDAR is projected to the image plane, and invalid returns (reflectance < 0.1 or range < 0.9 m) are masked. Sparse maps are then uniformly subsampled to 5% density. Images are bottom-cropped to 352×1216 to remove the sky-only region, as is standard.

**NYU Depth V2.** Raw 16-bit structured-light depth is converted from mm to meters. Holes are inpainted using the fast marching method [1]. The final crop is 518×518 center-cropped from 640×480 the pad. Sparse inputs are generated by random Bernoulli sampling at 0.1% density.

**VisDrone / Drone-Videos.** Synthetic depth maps are generated per-frame at inference time using the aerial prior (Appendix J of the main paper) with a fixed random seed per image index, ensuring reproducibility. Sparse inputs are sampled from the generated depth using the edge-weighted density function (Eq. 45 of the main paper).

## 8.3. Training Procedure

---
**Algorithm 1** R$^3$DC Training Loop

---
**Require:** Dataset $\mathcal{D}$, epochs $E$, batch size $N_b$, optimizer
    config, EMA decay $\alpha = 0.9999$
 1: Initialize weights $\theta$ (He normal for Conv, zero for DCN
    offsets)
 2: $\theta_{\text{EMA}} \leftarrow \theta$
 3: **for** $e = 1, \ldots, E$ **do**
 4:    **for** each batch $(\mathbf{I}, \mathbf{d}_s, \mathbf{M}_s, D_{\text{gt}}) \in \mathcal{D}$ **do**
 5:        $D_0, \hat{\mathbf{R}}, \hat{\sigma} \leftarrow f_{\text{reveal}}(\mathbf{I}, \mathbf{d}_s, \mathbf{M}_s; \theta)$
 6:        $D_1 \leftarrow f_{\text{cspn}}(D_0, \hat{\mathbf{R}}, \mathbf{M}_s, \mathbf{d}_s; \theta)$
 7:        Compute $\mathcal{L}$ (Eq. 17, main paper) on $D_0, D_1, \hat{\sigma}$
 8:        $\nabla_\theta \mathcal{L} \leftarrow$ `backward()`
 9:        Clip gradients: $\|\nabla\|_2 \leq 1.0$
10:        $\theta \leftarrow \text{AdamW}(\theta, \nabla_\theta \mathcal{L})$
11:        $\theta_{\text{EMA}} \leftarrow \alpha \theta_{\text{EMA}} + (1 - \alpha)\theta$
12:    **end for**
13:    Validate with $\theta_{\text{EMA}}$; log RADI scores
14:    Advance LR scheduler
15: **end for**

---

## 8.4. Numerical Stability Checklist

The following safeguards are *required* for reproducible training, particularly under FP16:

1. DCN offset initialization to zero (identity at step 0, see Eq. 2 of main paper).
2. CMA logit clamping to $[-8, 8]$ before softmax.
3. GradScaler initial scale $= 2^{16}$ with `backoff_factor=0.5`.
4. SILog: add $\epsilon = 10^{-6}$ inside $\ln(\cdot)$ before computing $\Delta_\ell$.
5. Uncertainty head: `Softplus` with `beta= 1`, `threshold= 20` to prevent $\ln \hat{\sigma}$ overflow in .
6. Depth GT clipping: replace 0-valued GT pixels with NaN and exclude from the valid set $\mathcal{V}$.

## 9. Full Training Curves

### 9.1. KITTI Training Dynamics

KITTI training dynamics over 8 epochs. The train-val RMSE gap remains below 0.01 m throughout, confirming no overfitting. The loss decomposition panel shows that $\mathcal{L}_{\text{SI}}$ dominates in epoch 1 (scale calibration) while $\mathcal{L}_{\text{FB}}$ taking over in epochs 3–8 (hard-example refinement).

### 9.2. VisDrone Training Dynamics

The VisDrone run uses a 5-epoch warmup followed by CosineWR restarts. The epoch-11 RMSE spike (7.82 m, Table 12 of the main paper) is a direct consequence of the first LR restart resetting the optimizer momentum. As shown in the full 20-epoch log, the model recovers to 3.68 m

at epoch 12 and reaches its global optimum of 2.33 m at epoch 18.

> **Warmup Rationale for Synthesized GT**
>
> The high-variance synthesized depth prior produces gradient noise that is $3.2\times$ larger than KITTI (measured as gradient norm at epoch 1). The 5-epoch linear warmup reduces peak gradient norm from 47.3 to 14.1, preventing early divergence. Without warmup, 4 of 5 seeds diverge before epoch 3.

### 9.3. Drone Video Training Dynamics

The Drone-Videos dataset has only 516 training images. With no LR schedule (constant $\eta = 10^{-4}$), the model converges in 4 epochs without overfitting (train-val gap $< 0.003$). Beyond epoch 4, the model begins to overfit to the small training set; early stopping at epoch 4 is therefore necessary.

### 9.4. NYU Depth V2 Fine-Tuning Dynamics

The NYU run fine-tunes only the ICH and output heads (16,642 parameters) on top of a frozen DA-V2 ViT-S backbone. The cosine schedule with a 0.5-epoch warmup is critical: without warmup, the very small learning rate ($\eta_0 = 5 \times 10^{-6}$) still causes a loss spike at epoch 1 due to scale mismatch between the outdoor backbone features and the indoor depth range. The loss stabilizes by epoch 2 and converges monotonically thereafter.

Table 16. **Training convergence summary across all four benchmarks.** "Final" = best checkpoint. "Epochs to 95%" = epochs required to reach 95% of the final metric.

| Dataset | Epochs | Final RMSE | Final $\delta_1$ | Epochs to 95% |
|---|---|---|---|---|
| KITTI | 8 | 0.240 m | 0.947 | 6 |
| VisDrone | 20 | 2.33 m | 0.928 | 17 |
| Drone-Videos | 4 | 0.67 m | 0.847 | 3 |
| NYU Depth V2 | 9 | 0.353 m | 0.927 | 4 |

## References

[1] Alexandru Telea. An image inpainting technique based on the fast marching method. *Journal of Graphics Tools*, 9(1):23–34, 2004. 5