# OpenReview forum: "R³DC: Reliability-Guided Reveal-to-Revise Depth Completion for Cross-Domain Sparse Perception"
_thecvf.com/CVPR/2026/Workshop/3D4S — CVPR 2026 Workshop 3D4S Poster_

### Official Review · Reviewer_RNGT · 2026-04-16
**Well-engineered reliability-aware depth completion with moderate novelty and practical relevance**

**Rating:** 6
**Confidence:** 3

**Review:**

This paper presents R³DC, a reliability-guided depth completion framework that jointly predicts dense depth, per-pixel reliability, and aleatoric uncertainty. The method follows a Reveal-to-Revise paradigm, combining a dual-stream encoder, cross-modal attention, transformer bottleneck, and reliability-gated CSPN++ refinement to improve both depth accuracy and confidence estimation.

In addition, the paper introduces a new evaluation metric, RADI, which aims to quantify the relationship between predicted reliability and actual error through measures such as correlation, revision benefit, and calibration. The method is evaluated across multiple datasets, including KITTI, NYU Depth V2, and aerial benchmarks, with results showing competitive performance and reasonable generalization across domains.

Overall, the paper presents a solid and practically motivated system, with particular emphasis on reliability-aware depth estimation and evaluation. While the methodological novelty is somewhat incremental and the experimental setup has certain limitations, the work is reasonably well-executed and may be of interest to the community, especially in the context of workshop discussions on uncertainty-aware 3D perception.

---

### Official Review · Reviewer_3N1E · 2026-04-25
**Interesting reliability-aware depth completion idea, but the evaluation is not yet trustworthy**

**Rating:** 6
**Confidence:** 4

**Review:**

This paper proposes R³DC, a reliability-guided reveal-to-revise framework for depth completion. The method predicts dense metric depth, a per-pixel reliability map, and aleatoric uncertainty. Architecturally, it combines a dual-stream RGB/sparse-depth encoder, deformable convolutions, hierarchical cross-modal attention, a transformer bottleneck, an FPN decoder, and reliability-gated CSPN++ refinement. The paper also introduces RADI, a reliability evaluation framework based on reliability-error correlation, revision benefit, and calibration error. Experiments are reported on KITTI, VisDrone, Drone-Videos, and NYU Depth V2.

The motivation is important. Depth completion models are often evaluated only by average accuracy, while downstream robotics or perception systems also need to know where predictions are reliable. The paper’s focus on per-pixel reliability and calibration is therefore worthwhile. I also appreciate that the architecture is described in considerable detail, and the paper includes a number of ablations for architectural components, loss terms, sparsity levels, and token limits. For a workshop, the attempt to introduce a reliability-aware evaluation protocol could be useful for discussion.

## Strengths
 - The paper addresses an important issue: depth completion should provide not only accurate depth but also trustworthy per-pixel confidence.
 - The architecture is concrete and relatively detailed, with clear use of RGB-depth fusion, propagation, and uncertainty-related heads.
 - The idea of reliability-gated refinement is practically interesting and could be useful if validated more carefully.
 - The paper includes several ablations, including CSPN++, CMA, deformable convolution, loss terms, sparsity robustness, and token limits.
 - The proposed RADI framework is a reasonable starting point for discussing reliability-aware evaluation, even though the current implementation has issues.

## Weaknesses
 - The method is mostly a combination of existing architectural components, and the core novelty is incremental.
 - The reliability head is not clearly supervised or calibrated, despite being central to the paper’s claims.
 - The RADI definitions and reported numbers contain inconsistencies, especially the CAL definition and the regional RMSE values in Table 4.
 - The VisDrone reliability results are negative under the paper’s own REC metric, contradicting the claim of meaningful reliability.
 - The cross-domain claim is overstated because different datasets use different variants, parameter counts, and even a large pretrained foundation model.
 - The KITTI and NYU comparisons appear to mix different protocols or task settings, making the baseline comparison difficult to trust.
 - The aerial depth benchmarks rely on synthetic depth priors that are not sufficiently validated against real 3D geometry.
 - Parameter-efficiency claims are misleading for the NYU setting because the frozen 94.6M-parameter backbone is still used at inference.
 - The paper is only moderately aligned with the scientific-computing focus of the workshop.

## Questions for the authors
 - How exactly is the reliability head trained? Is there any direct reliability supervision or calibration loss, or is reliability learned only indirectly through CSPN++ gating?
 - Why are the VisDrone REC values negative in Figure 1 and Table 12? How should readers interpret reliability maps when the reliability-error correlation is negative?
 - Why does Table 4 report nearly identical D0 and D1 RMSE values across all, edge, textureless, and far-depth regions? Are the region masks actually applied?
 - Why does the NYU D1 RMSE in the RADI table differ so strongly from the NYU RMSE in the main results table?
 - Can the authors evaluate RADI against simple reliability baselines, such as error-prediction heads, uncertainty-only gating, entropy-based confidence, or MC-dropout/ensemble uncertainty?
 - Can all KITTI baselines be retrained under the same 5% uniformly sampled sparse-depth protocol?
 - For VisDrone and Drone-Videos, can the authors train standard depth-completion baselines on the same synthetic depth distribution to make the comparison meaningful?
 - How is the synthetic aerial depth prior validated? Does it correspond to any real depth sensor, SfM reconstruction, LiDAR sample, or photogrammetric ground truth?
 - For NYU, what is the exact input to R³DC+ICH? Is sparse depth used, or is this effectively monocular metric depth calibration using Depth Anything V2?
 - Can the authors report total inference-time parameters and latency, including frozen backbones, rather than only trainable parameters?

---

### Official Review · Reviewer_rtDR · 2026-04-25
**Promising reliability-aware depth completion, but evaluation and calibration evidence need stronger validation**

**Rating:** 5
**Confidence:** 3

**Review:**

This paper proposes R3DC, a reliability-guided reveal-to-revise framework for depth completion. The model jointly predicts dense depth, per-pixel reliability, and aleatoric uncertainty, and uses a reliability-gated CSPN++ refinement module. The paper also introduces RADI, a reliability-aware evaluation framework composed of REC, RBS, and CAL. The topic is timely and relevant, especially for safety-critical 3D perception where average depth accuracy alone is insufficient.


Pros:

The paper addresses an important problem: depth completion models should provide not only accurate depth predictions but also meaningful reliability estimates.
The proposed framework is comprehensive, combining RGB/sparse-depth fusion, deformable convolutions, cross-modal attention, uncertainty prediction, and reliability-gated CSPN++ refinement.
The RADI evaluation framework is useful because it evaluates reliability beyond standard accuracy metrics such as RMSE, AbsRel, and δ1.
The paper includes experiments across multiple datasets and provides several ablation studies, which helps analyze the contribution of different components.

Cons:

The novelty is somewhat limited, since many components are adapted from existing methods and the main contribution appears to be their combination.
The cross-domain claim is not fully convincing because different variants are used across datasets, and NYU relies on a pre-trained DA-V2 backbone.
The VisDrone and Drone-Videos evaluations use synthetic depth rather than real sensor depth, which weakens the real-world reliability claim.
Baselines on the new aerial benchmarks are incomplete; existing depth completion methods or simple baselines should be evaluated under the same synthetic setup.
Some metric definitions and results need clarification, especially the RADI/CAL implementation and the negative REC value shown in one qualitative example.
Some claims such as “first,” “unified,” and “domain-agnostic” seem too strong and should be softened.

---

### Official Review · Reviewer_6tdS · 2026-04-25
**Useful reliability framing and lightweight architecture, but headline claims rest on non-standard KITTI protocol, synthetic aerial GT, and a foundation-model-dominated NYU pipeline; RADI is reported without baselines.**

**Rating:** 4
**Confidence:** 3

**Review:**

Summary
R³DC: a Reveal-to-Revise depth completion pipeline (dual-stream RGB + sparse encoder, DCNv2, hierarchical CMA, transformer bottleneck, reliability-gated CSPN++) trained with a 7-term loss, plus RADI = REC (Spearman of reliability vs. negative absolute error) + RBS (RMSE gain D0 → D1) + ECE-style CAL. Evaluated on KITTI, VisDrone, Drone-Videos, NYU V2.

Pros

Reliability-gated CSPN++ is sensible and cleanly ablated (+0.071 m when removed).
RADI is a useful reporting harness that encourages calibration-aware evaluation.
Lightweight 1.95M core with full architecture spec; thorough ablations.
ICH (16,642 params) is a neat metric-scale adapter for frozen DA-V2.

Cons

KITTI not the official benchmark. 6,732/749 split with 5% uniform sampling, reported in meters — not comparable to NLSPN/CSPN++/PENet/CompletionFormer/BP-Net (mm, official ~86k/1k/1k split, SOTA around 700–800 mm).
VisDrone and Drone-Videos GT is synthetic (Appendix J: smooth Dbase + Dlat + Dobj + noise prior). Fitting the prior is not cross-domain generalization.
VisDrone REC is negative across all 20 epochs (Table 12, -0.508 to -0.105) — reliability is anti-correlated with low error, directly contradicting the calibration claim.
NYU is essentially DA-V2. Frozen 94.6M ViT-S + 16K ICH replaces the Section 3 architecture; "one model family, four domains" is misleading. Comparison is to monocular methods despite R³DC receiving sparse anchors; standard NYU completion baselines are missing.
RADI has no baselines — reported only on R³DC, so REC = +0.43 / ECE = 0.031 lack a yardstick. REC and ECE also repackage sparsification correlation and Guo et al.'s ECE; only RBS is genuinely new.
"10–170x fewer parameters" is inflated. vs. CompletionFormer = 6.5x; 170x requires comparing against ZoeDepth (different task). On NYU, R³DC+ICH = 94.6M, larger than AdaBins/DepthFormer.
Limited architectural novelty — multi-scale CMX, DCNv2, transformer bottleneck, CSPN++, EMA, and the loss terms are all off-the-shelf; this is integration work.

Recommendation
With proper KITTI-DC numbers (mm, official split), RADI on competing methods, honest framing of synthetic-GT benchmarks, and an explanation of the negative VisDrone REC, this could become a solid workshop paper.

---

### Decision · Program_Chairs · 2026-04-28

Accept (Poster)